# Dynamics and Impacts of Monsoon-Induced Geological Hazards: A 2022 Flood Study along the Swat River in Pakistan

Nazir Ahmed Bazai[1,3], Mehtab Alam[4,5]**, Peng Cui [1,2,3]*, Wang Hao[1], Adil Poshad Khan[5], Muhammad Waseem[5], Yao Shunyu[6], Muhammad Ramzan[1], Li Wanhong[1,3,] Tashfain Ahmed[7],

[1] Key Laboratory of Mountain Surface Process and Hazards/Institute of Mountain Hazards and Environment, Chinese Academy of Sciences, Chengdu, 610041, China.

[2] Key Laboratory of Land Surface Pattern and Simulation/Institute of Geographic Sciences and Natural Resources Research, Chinese Academy of Sciences, Beijing 100101, China.

[3] China-Pakistan Joint Research Center on Earth Sciences, CAS-HEC, Islamabad 45320, Pakistan.

[4] Department of Geotechnical Engineering, College of Civil Engineering, Tongji University, Shanghai 200092, China.

[5] Ghulam Ishaq Khan Institute of Engineering Sciences and Technology, Topi 23640, District Swabi, Khyber Pakhtunkhwa, Pakistan.

[6] China Institute of Water Resources and Hydropower Research, Beijing 100038, China.

[7] Deep Tech Lab, Computer Science and Engineering Department, Michigan State University, East Lansing, Michigan 48823, USA.

*Correspondence to: Peng Cui *pengcui@imde.ac.cn and Mehtab Alam** mehtab.alam@giki.edu.pk*

## Abstract

This study examines the impacts of the unprecedented 2022 monsoon season in Pakistan's Swat River basin, where rainfall exceeded historical averages by 7-8%. This extreme weather led to catastrophic debris flows and floods, worsening challenges for low-income communities. The resulting financial instability affected millions, causing significant damage to homes, crops, and transportation. The study employs a multidisciplinary approach, combining field investigations, remote sensing data interpretation, and numerical simulations to identify the factors contributing to debris flow incidents. Analysis of land cover changes reveals a decrease in grasslands and an increase in barren land, indicating the adverse effects of deforestation on the region. Topography and gully morphology are crucial in initiating debris flows, with steep gradients and shallow slope failures predominant. Numerical simulations show that debris flows reached high velocities of 18 m/s and depths of 40 m within 45 minutes. Two debris flows resulted in the formation of dams along the Swat River, intensifying subsequent floods. The study emphasizes the interplay of extreme rainfall and deforestation during the rainy season, rendering the region susceptible to debris flows and hindering restoration efforts. Recommendations include climate change mitigation, reforestation initiatives, and discouraging construction activities in flood-prone and debris-flow-prone regions. The study advocates for enhanced early warning systems and rigorous land use planning to protect the environment and local communities, highlighting the imperative of proactive measures in the face of escalating climate challenges. Additionally, the study investigates the spatial distribution of various events and their consequences, including potential hydro-meteorological triggers, and how such events initiate processes that change mountain landscapes. It also assesses the extent to which the 2022 monsoon can be classified as abnormal. The combination of empirical evidence and practical insights presented in this study highlights research gaps and proposes routes toward attaining a comprehensive comprehension of monsoon-triggered geological hazards and consequences.

**Keywords:** Climate Change; Geological Hazards; Monsoon Floods; Deforestation, Future Challenges

## 1. Introduction

Disasters like floods, earthquakes, droughts, heat waves, tsunamis, cyclones, etc., devastate human systems, killing thousands of people and destroying infrastructures worldwide, causing billions of economic losses (O'brien et al., 2008; Ward et al., 2020). The impact of these disasters fluctuates from country to country depending upon the socioeconomic resilience and geomorphology of the land (Atta-Ur-Rahman, 2010). In developed nations, the economic impact of extreme weather events tends to be greater than in developing countries. Conversely, developing countries experience a higher number of human casualties associated with these events (Atta-Ur-Rahman, 2010). Developing countries are particularly vulnerable to catastrophic events, often serving as hot spots for such occurrences. This heightened risk is largely attributed to unplanned and rapid infrastructure development, which frequently lacks adequate resilience to withstand natural hazards. Furthermore, limited resources for disaster preparedness and response amplify the impacts of these events, leaving communities more exposed to potential devastation (Mayhorn and Mclaughlin, 2014). The Centre for Research on the Epidemiology of Disasters (CRED (2014) estimation shows that the frequency of disasters increased from 100 disasters per decade from 1900 to 1940, whereas 2080 disasters were extreme from 1900 to 2000.

In 2013, Asia was the third hardest-hit region, accounting for about 88% of global disaster-related casualties—significantly higher than the decadal average of 62%. In the same year, Pakistan was classified as fifth-ranked among the most affected countries by CRED (2014), whereas German Watched ranked Pakistan as the third among highly affected nations (Kreft et al., 2013). The consequences of global warming are becoming more and more feasible and practical in the form of increasing natural floods and putting people at risk in several ways, mainly in South Asian regions (Shah et al., 2017). Since 1959, Pakistan has contributed only 0.4% of global carbon dioxide emissions—the primary greenhouse gas—compared to 16.4% from China and 21.5% from the United States (Handley, 2022). Yet, Pakistan remains among the countries most severely impacted by catastrophic mega-floods, experiencing these devastating events nearly every decade.

Climate change has increased the intensity and frequency of hydrometeorology events that are always a significant threat to water, food, energy, infrastructure, and human lives yearly (Moghim, 2018). Temperature changes and highly increased precipitation, especially in the monsoon season, are significant disaster factors in Pakistan (Otto et al., 2023; Hussain et al., 2024). Monsoons and heavy rainfall are the main factors inducing natural floods and geological hazards (Segoni et al., 2018; Amarasinghe et al., 2024). Flash floods and debris flow dominate during heavy rainfall in the mountainous region. The flash flood, characterized by rapid, short-duration, and high-velocity flows, is the primary cause of property damage and casualties (Ma et al., 2020). According to the "NWS Glossary," flash floods are rapid water rises in "a stream or creek above a predetermined flood level, beginning within six hours of the causative event" (Nws, 2009). Flash floods are mainly caused by short-duration and high-intensity rainfalls in watersheds smaller than 260 km$^2$ (Davis, 2001; Georgakakos and Hudlow, 1984; Tang et al., 2020). In this article, flash floods include river floods, rainfall-induced landslides, and debris flows. Flash

floods are one of the world's most devastating and fatal disasters (Špitalar et al., 2014; Tang et al., 2017; Jonkman et al., 2024).

The flooding in Pakistan and surrounding regions in recent decades has demonstrated its severity and extremes (Gul et al., 2024). The most recent and most destructive was the unprecedented monsoon rains in 2022 from June to September triggered one of Pakistan's worst floods in decades, and the Pakistani government declared a state of emergency on August 25th, according to Pakistan's National Disaster Management Authority (NDMA). The floods have led to over 1,718 casualties and 12,800 injuries, with the highest fatality rate in Sindh. More than 33 million people (about 15% of the country's total population) were affected by the floods in 84 calamity-declared districts, some 7.9 million people were displaced, and some 598,000 lived in relief camps. Over 287,000 houses have been destroyed, and over 662,000 are partially damaged (NDMA). At the same time, the flood caused damage to roads, railway tracks, telecommunication systems, and other infrastructure to varying degrees, seriously affecting the lives of residents. This disturbance to infrastructure, residences, and livelihoods continues to worsen over time. The situation has been exacerbated by a lack of timely response and identification of climatic factors, inadequate response from residents and local authorities, and the absence of timely implementation of proposed strategies (Shah, 2020; Mani and Goniewicz, 2023; Xu et al., 2024).

Recent human-induced changes significantly influence the likelihood of slope movements in how people use the land (Vanacker et al., 2003; Reichenbach et al., 2014; Liu et al., 2024) and treat the surroundings of the rivers. Changing from forest land-cover/land-use (LCLU) classes can increase the vulnerability to landslides and intense runoff. Including vegetation enhances slopes' physical stability by augmenting root-soil strength (Hürlimann et al., 2022; Mehtab et al., 2020; Bazai et al., 2021). It reduces erosive forces through the canopy's interception of rainfall drops (Bonnesoeur et al., 2019). Furthermore, slope vegetation has been documented to decrease infiltrating water and lower pore water pressure (Tsukamoto, 1990; Sidle, 1992; Preti et al., 2010). On the contrary, it may also reduce infiltration by encouraging soil crust formation (Bu et al., 2014). Since 1850, the planet has witnessed an alarming and unprecedented destruction of forests, driven mainly by the expansion of industrial capitalism and resulting in a global forest extinction crisis. Forests in the mountainous regions of Pakistan are undergoing rapid degradation, particularly in these areas. The deforestation rate stands at nearly 1.5% (Mehmood et al., 2018), a highly concerning and alarming trend posing a significant threat to the ecosystem, which also often leads to geological hazards and flash flooding in the mountainous region of Pakistan (Pandey et al., 2022; Ansari et al., 2024).

The different factors contributing to geological hazards have been well identified, but current studies have primarily focused on isolated factors and their potential consequences. However, there remains a gap in research for a comprehensive approach that considers the interplay between extreme climate events and geological hazards, incorporating various triggering factors, the intensity of debris flow, and their resulting consequences. This study addresses this gap by examining the combined effects of multiple triggering factors on the frequency and intensity of debris flows, their role in triggering secondary events such as floods, and the subsequent infrastructural damage. Additionally, it considers how human activities worsen vulnerability, heightening both economic and life risks. To frame the study's objectives, this focused on understanding the

conditions under which debris flows occur, particularly in relation to rainfall thresholds, and the effects of deforestation as identified through both field observations and numerical modelling. Lastly, this study offers recommendations for mitigation strategies, aiming to support government authorities in implementing proactive measures to prevent future hazards. This comprehensive approach offers insights that could enhance hazard prediction, early warning systems, and resilience planning in affected regions.

## 2. Study Area

The Swat District in northern Pakistan (Fig. 1) spans a complex topographical and hydrological region centered on the Swat River. The river originates from the Hindu Kush Mountains, at an elevation of around 3,000 meters, with year-round contributions from glaciers in the Usho and Gabral valleys. These glaciers are crucial for maintaining river discharge, particularly during the warmer months (June to September) when melting intensifies, resulting in increased flow alongside
the seasonal monsoon rainfall from July to September (Anjum et al., 2016; Dahri et al., 2011).

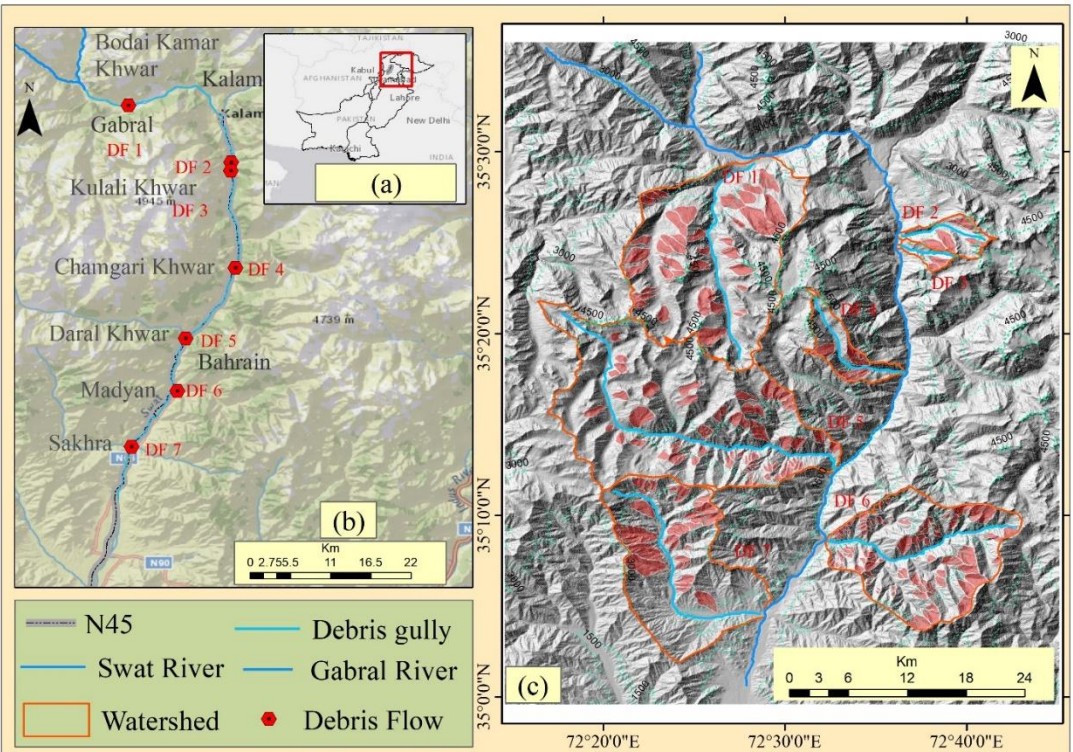

**Figure 1.** Location and details of the study area. (**a**) Study area; (**b**) Details of the Swat River area and its investigated debris flow; (**c**) Catchments of debris flow fans, loose material, and landslides used as the primary source of debris flow. The source of the background image is © Google Earth.

At Kalam, where the Usho and Gabral rivers converge, the Swat River flows through high valleys with steep gradients, especially within narrow gorges in the Kalam Valley. These gradients increase flow velocity, which heightens risks of flash flooding, erosion, and debris flows in the upper catchments. Upon reaching Madyan, the valley broadens, and the river's gradient decreases, which moderates flow speed but still presents flood risks, especially when sediment deposits alter the river course (Rahman et al., 2023). Downstream, the river flows over the plains of the lower Swat Valley, covering

approximately 160 km to Chakdara, where gentler slopes and broader valley floors reduce flow speed but create vulnerability to overbank flooding during heavy rains or rapid snowmelt.

Further downstream, the Swat River narrows again as it meets the Panjkora River near Bosaq before entering the Peshawar Valley, ultimately joining the Kabul River near Charsadda. This confluence in confined spaces increases the potential for flooding due to the greater water volume and restricted flow paths. The elevation variation, from the mountainous upper

reaches to the lower plains, affects the river's hydrological behavior, with steeper gradients supporting high-energy flows that carry sediment and may cause landslides, while lower gradients promote sediment deposition and increase floodplain areas at risk of flooding (Anjum et al., 2016; Dahri et al., 2011).

Recent studies have raised concerns about the impact of climate change on the Swat River's flow patterns, particularly regarding glacial retreat. The glacial retreat could affect seasonal flows and potentially amplify flooding risks during

monsoon seasons as glacial meltwater decreases over time (Rahman et al., 2023).

## 3.    Methodology

### 3.1 Field Investigation

In the last few decades, the northern region of Pakistan, particularly Swat, has experienced a significant number of geological hazards, especially during the monsoon season. As a result of these occurrences, roads often become impassable

due to various geological hazards such as landslides and debris flows (Islam et al., 2022b). In the 2022 monsoon season, heavy rainfall led to several landslides and debris flows, causing casualties that were extensively covered by the media. After consulting different local government bodies of Pakistan, i.e., the Communication and Works Department and Provisional Disaster Management Authorities (PDMA), the various locations of catastrophic geological hazards were identified through extensive field visits. The sites were then physically inspected to study the root cause of the identified geological risks. The

predominant geological hazards observed in the area were landslides and debris flows (DF). The authors carefully documented the location of each landslide and debris flow using GPS, with an accuracy of 1m. To ensure the precision of the GPS, it was calibrated against established benchmarks. The accuracy was further verified by comparing GPS data with high-resolution satellite imagery. Additionally, multiple GPS devices were used to record the location of the same point, enhancing data reliability by providing redundant measurements for cross-checking and minimizing potential errors. The

main challenges encountered in the rugged environment included satellite signal obstruction, multipath errors, terrain-

induced errors, battery limitations, and adverse climatic conditions, all of which can impact GPS accuracy and data consistency. These challenges were addressed through strategies such as calibration against benchmarks, cross-verification with satellite imagery, ensuring optimal device maintenance, and employing backup technologies like GIS to enhance data accuracy and reliability. They also utilized measuring tape to determine the dimensions of these landslides. Furthermore, the

dimensions of each debris flow dam were also measured and documented comprehensively. The data collected from the field visits were then analyzed to understand the features of the debris flow source areas, the flow paths, and flow deposits in the study region. The gully of each identified debris flow was surveyed on the ground, and understanding was gained about defining the source conditions and the characteristics of the flow channel and the depositional fan. The identified geological hazards were mapped using a basic geographical information system (GIS). After an extensive survey of the debris flow

gully, the damage assessment of structures in the study area was performed. The debris flows were categorized into different intensities, namely low, medium, high, and very high, based on the extent of damage they caused and the volume of debris involved.

### 3.2 Metrological and Remote Sensing Data

Rainfall records from the meteorological station near Kalam, Swat (Lat = $35^0 50'$, Long. = $72^0 59'$) were used to determine

the daily rainfall totals during all the debris flow events. Hourly precipitation data collected at the rain gauge station were used to determine the rainfall intensity/duration for each debris flow-producing rainstorm. The precipitation data were collected from the Meteorological Department of Pakistan (PMD). On August 26, 2022, the Irrigation Department of Khyber Pakhtunkhwa (KPK) reported a record-breaking flood discharge of 227,899 cusecs from the River Swat, creating a hazardous situation for the residents in its vicinity.

The satellite data include Land Use and Land Cover (LUC) and a Geographic Information System-derived Digital Elevation Model (DEM). They were obtained from sites such as NASA's Earth Explorer and other trusted sources like the US Geological Survey (USGS) and Sentinel-2. To enhance spatial comparability, all the datasets were resampled at a 30-meter grid cell size.

### 3.3 Land Use and Land Cover (LUC)


Various types of Land Use and Land Cover (LUC) can impact the stability of slopes due to their ability to alter the hydrological processes of hillslopes, the distribution of rainfall, the characteristics of infiltration, runoff generation, and even the shear strength of the soil itself (García-Ruiz et al., 2010). However, unlike numerous environmental factors like geological structure and lithology, Land Use and Land Cover (LUC) have the potential to undergo seasonal or rapid changes

due to the combined effects of natural processes and human activities (Reichenbach et al., 2014)). Therefore, in regions characterized by rapid fluctuations in Land Use and Land Cover (LUC) within a short timeframe, it becomes crucial to investigate the influence of LUC on landslides (Ali et al., 2012). A comparative analysis spanning at least two distinct periods is necessary for LUC mapping to capture the LUC changes, as Pisano et al. (2017) outlined. In addition to examining historical LUC, the potential impact of changes in land use, such as urbanization and deforestation, was considered

qualitatively in relation to slope stability and debris flow risks. The present study examined 20 years (2002–2022), segmented into 04 intervals: 2002, 2009, 2016, and 2022. The analysis utilized MODIS MCD12Q1 V6.1 data obtained from the USGS Earth Explorer platform (https://earthexplorer.usgs.gov/) and involved a comprehensive workflow executed in ArcGIS. The land cover classification was performed on each year's dataset using ArcGIS tools. This involved applying appropriate spectral indices, thresholds, and classification algorithms to assign specific land cover categories to each pixel.

Change detection analysis was subsequently conducted by comparing classified maps for consecutive years. Changes in land cover percentages were quantified for each type. Temporal trends in land cover were analyzed by calculating the percentage distribution of each land cover class across the study period. Emphasis was placed on identifying persistent patterns and notable shifts in vegetation, croplands, barren land, built-up areas, and other land cover types. The results were visualized using thematic maps to illustrate the evolving LUC patterns.

**3.4 Governing equations of a numerical method**

The 2-D debris flow movement model that simplifies the Navier–Stokes equation via the depth-integrated continuum method was adopted to analyze the dynamic processes of the 04 (DF 2, DF 3, DF 4, and Df 5) strong debris flows induced by landslides and loose material by heavy rainfall in the catchments of the study area. In this context, the study also considered the effects of varying Land Use and Land Cover (LUC) on the debris flow dynamics. Changes in LUC, such as deforestation

and urbanization, could alter runoff patterns and the volume of material available for debris flow, impacting the flow dynamics. The governing equations are as follows:

$$\frac{\partial W}{\partial t} + \frac{\partial F}{\partial x} + \frac{\partial G}{\partial y} = S \qquad (1)$$

in which

$$W = \begin{bmatrix} h \\ hu \\ hv \end{bmatrix}, F = \begin{bmatrix} hu \\ hu^2 + gh^2/2 \\ huv \end{bmatrix}, G = \begin{bmatrix} hu \\ huv \\ hv^2 + gh^2/2 \end{bmatrix}, S = \begin{bmatrix} 0 \\ gh(S_{ax} - S_{fx}) \\ gh(S_{ay} - S_{fy}) \end{bmatrix} \qquad (2)$$

In this context, h represents the flow height, while $u$ and $v$ denotes the depth-integrated flow velocities in the x and y directions, respectively. The parameter g stands for the acceleration due to gravity, $S_{ax}$ and refers to the momentum source terms in the $x$ and $y$ directions, respectively. Additionally, $S_{fx}$ and $S_{fy}$ are the resistance terms in the $x$ and $y$ directions, respectively. The format of $S_{ax}$ and $S_{ay}$ is as follows:

$$S_{ax} = -\frac{\partial Z}{\partial x}, S_{ay} = -\frac{\partial Z}{\partial y} \qquad (3)$$

The field investigations revealed that the debris flows were viscous. In debris flow simulations, a viscous debris flow is usually simplified to the Bingham fluid model (Julien and Lan, 1991; Rickenmann et al., 2006; Ying-Hsin et al., 2013).

Therefore, the Bingham fluid stress constitutive model was used as the resistance model. Considering the impact of changing LUC, the model accounted for variable soil cohesion and friction factors, which might differ depending on the land cover type (e.g., forested vs. urbanized areas). Initial conditions for the debris flow simulations were defined based on field data from source zones, including measured flow depth and velocity at t = 0 min to ensure that the starting parameters accurately reflected real conditions. Parameters such as channel slope, sediment concentration, and water flow rates were calibrated using local hydrological data to enhance simulation fidelity. Furthermore, the simulation outputs were validated against field observations and past debris flow events, ensuring a close alignment between observed and simulated values for flow velocities and depths. This model calibration and validation approach strengthens confidence in the simulations' accuracy and reliability, effectively capturing the dynamics of the debris flows observed in the Swat River basin.

The total Bingham friction $S_f$ is as follows:

$$S_f = \frac{\tau_{\mathrm{B}}}{\rho g h} + \frac{K_l \mu_{\mathrm{B}} V}{8 \rho g h^2} + \frac{n^2 V^2}{h^{4/3}} \qquad (4)$$

where $\tau_{\mathrm{B}}$ is the Bingham yield stress; $\mu_{\mathrm{B}}$ is the Bingham viscosity; $K_l$ is the laminar flow resistance coefficient; $n$ is the pseudo-Manning's resistance coefficient; $V$ is the depth-averaged velocity, i.e., $\sqrt{u^2 + v^2}$; and $\rho$ is the fluid density.

The relationship between $S_{fx}$ and $S_{fy}$, $S_f$, is as follows:

$$S_{fx} = \frac{u}{\sqrt{u^2 + v^2}} S_f, S_{fy} = \frac{v}{\sqrt{u^2 + v^2}} S_f \qquad (5)$$

The Massflow software (Ouyang et al., 2013) was used to solve the depth-integrated continuum governing equation. Since the Bingham resistance model is not defined in Massflow, a custom Bingham fluid resistance simulation was performed for the secondary development in Massflow. This simulation was further refined by incorporating LUC changes, which affect the material properties and flow resistance, ensuring that the model reflected realistic scenarios in areas with rapid land use changes. Experimental benchmarks and simulations of actual events have verified its robustness. It has been widely used in various two-dimensional surface flow simulation cases (Ouyang et al., 2019b; Ouyang et al., 2015a; Ouyang et al., 2015b; Ouyang et al., 2013; Ouyang et al., 2019a; Ouyang et al., 2016).

## 4. Results

### 4.1. Triggering Rainfall

Rainfall is recognized as one of the primary triggers for landslides and debris flows, especially in mountainous areas with heightened susceptibility. Previous studies, such as Cepeda et al. (2010), have highlighted the correlation between rainfall thresholds and increased landslide risks, supporting a threshold-based approach to landslide susceptibility assessment. Our

study found that intense rainfall events in the Swat River region were key triggers for debris flows, particularly when combined with critical antecedent rainfall conditions that heightened slope instability.

Hourly rainfall data from a gauge station at Kalam Valley (elevation 2,000 m) provided a reliable dataset to evaluate the precipitation characteristics associated with debris flow events in the study area. Specifically, both rainfall intensity and antecedent rainfall were quantified to identify their roles in triggering debris flows. The average monthly rainfall in the study area is shown in Fig. 2a, while Fig. 2b depicts the average daily rainfall for August 2022, a period of extreme rainfall. On August 26, 2022, intense rainfall of 71.5 mm/day was recorded, following a 40 mm cumulative rainfall over the two preceding days. This combination met the critical cumulative rainfall threshold, likely triggering multiple debris flows along the Swat River and resulting in significant structural damage throughout the affected area.

The recorded flood discharge across various locations is illustrated in Fig. 2c, where the Munda Dam experienced the highest discharge, attributed to the confluence of the Swat and Panjkora Rivers near Bosaq. The return period for this flood is estimated at 425 years, underscoring the event's rarity and severity. This extreme return period has critical implications for regional flood risk assessment, indicating that flood risk models must integrate rare, high-magnitude events to assess potential hazards effectively. The flood discharge was first reported on August 22, 2022, at Khwazakhela, and subsequently on August 26 at both Amandara Headworks (Batkhela) and Munda Dam (Mohmand Agency), reflecting the widespread impact of this event throughout the Swat River basin.

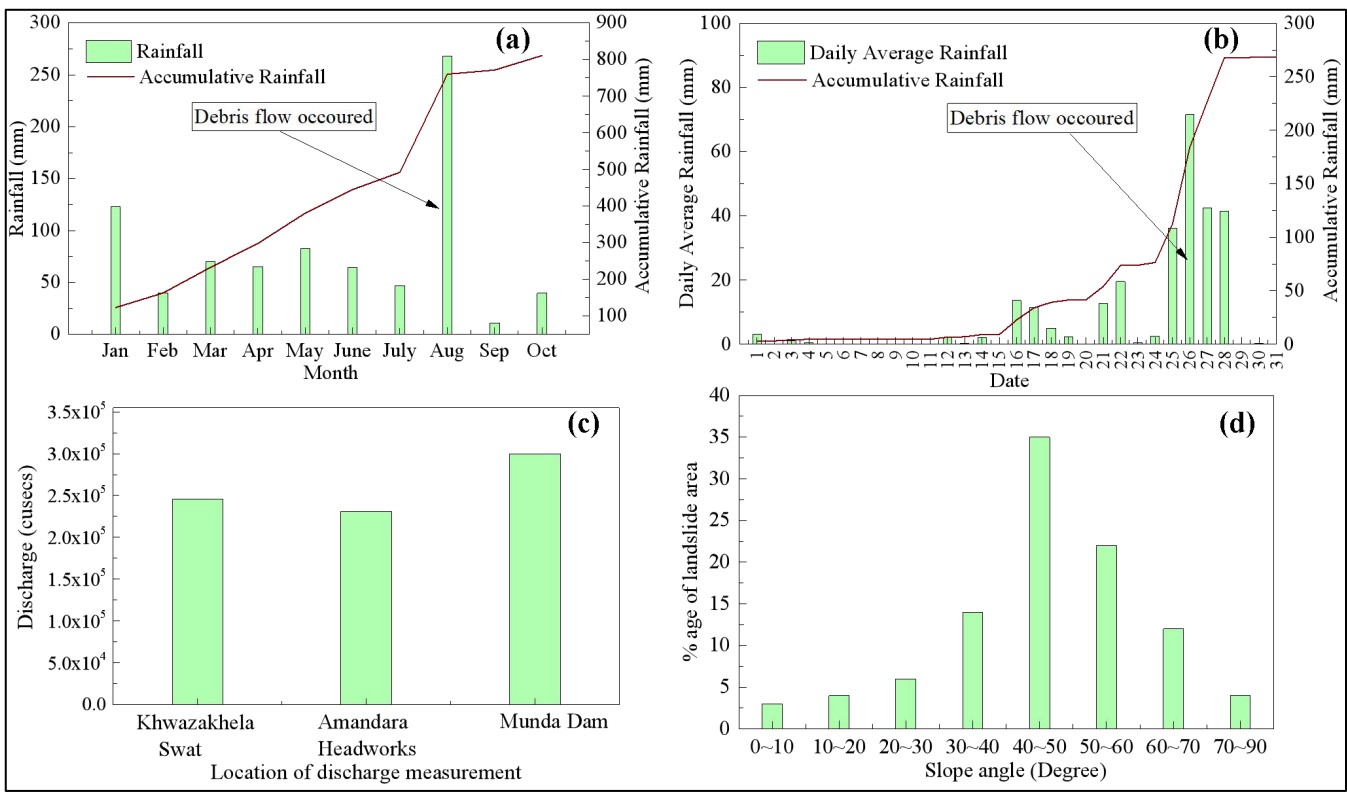

**Figure 2.** Precipitation data for the meteorological station at Kalam for the year 2022 (**a & b**), Precipitation data for the meteorological station at Kalam for August 2022 (**c**), Discharge measured at different locations along the Swat River during Flood 2022 (**d**), Landslide density in an area with varying angles of slope

### 4.2. Flood Level at different locations

The flood levels at various points along the Swat River and its tributaries were measured using hydrological gauges installed at critical points in the river's course and also through flood levels at open channels and flood-through marks (Table 01). In constrained areas with substantial residential encroachments, such as Kalam, Bahrain, and Madyan, flood levels reached between 50 to 80 feet. In contrast, in wider, less encroached sections of the river, such as Chakdara and Batkhela, flood levels ranged from 35 to 50 feet. The most extreme flood levels were recorded at Bodai Kamar Khwar (70-80 feet), with levels peaking at 95 to 105 feet at the confluence of the Swat and Panjkora Rivers. These high levels at confluence points significantly contributed to widespread residential inundation and damage in downstream areas.

**Table 1.** Flood Level observed at different locations along the Swat River during Flood 2022

| Location | Flood Level (ft) |
|---|---|
| Gabriel (Starting point of Swat River) | 40 |
| Kalam | 50-60 |
| Bahrain | 70-80 |
| Madyan | 70-75 |
| Khwazakhela | 45-50 |

| | |
|---|---|
| Chakdara (at Chakdara Bridge) | 30 |
| Batkhela (at Amandara Headworks) | 25 |
| Malakand Agency | 50 |
| Bosaq *(Junction of Swat and Panjkora Rivers)* | 95 -105 |
| Munda dam | 70 |
| Dagai (District Charsada) | 40-45 |
| Nowshera *(Kabul River)* | 25-30 |
| **Flood Level in Swat Tributaries** | |
| Bodai Kamar Khwar | 50 |
| Chamgari Khwar | 60 |
| Bodai Kamar Khwar | 70-80 |
| Daral Khwar | 40 |


## 4.3. Debris flow occurrence influenced by topographic and geological factors

The correlation between the volume of debris materials contributed by landslides and the total magnitude of debris flow events strongly depends on the topography and lithology of the gulley (Chen et al., 2024b). Typically, areas with steeper slopes and more significant elevations exhibit increased vulnerability to landslide and debris flow occurrences. Based on the
data extracted from the 25-m digital elevation model, it was observed that debris flows predominantly originated from steep slopes within relatively small drainage gulleys. As shown in Table 2, the average slope angles within the debris flow initiation zones ranged from 30° to 45°. The data depicted in Fig. 2d reveals that 87% of the landslides triggered by rainfall occurred in areas with slope angles exceeding 30°, whereas 83% occurred within the range of slope angles between 40° and 70°. The significance of slope steepness cannot be overstated when it comes to influencing the incidence of debris flows.
This occurs primarily because most debris flows begin as shallow landslides or as rill or gully erosion on large landslide deposits. In general, it can be concluded that debris flows are prone to be triggered on slope angles exceeding 30°.

Similarly, the lithological units of the region play a crucial role in the occurrence of debris flows. The geological units in the study area range from Mesozoic sedimentary rocks to Paleozoic and Precambrian igneous and metamorphic formations. DF1 and DF2 occurred in areas underlain by Mesozoic sedimentary rocks, such as sandstones and shales. These sedimentary
rocks are more prone to weathering and erosion due to their layered structures, where bedding planes act as planes of weakness. Specifically, shales become highly unstable when saturated and generally exhibit the highest rates of landslide activity, contributing significant amounts of loose material to debris flows.

In contrast, debris flows DF3–DF7 occurred in areas dominated by Paleozoic and Precambrian igneous and metamorphic rocks, including schists, marbles, and amphibolites. Although these rocks generally resist weathering, schists possess
foliation that creates planes of weakness. Additionally, fracturing and jointing caused by tectonic activity and past deformation weaken their internal structure, making these formations prone to failure. While amphibolites and marbles are more competent, they can still contribute to debris flows when fractured. These geological variations influence not only the triggering of debris flows but also the nature of the debris material. Sedimentary rocks provide finer-grained, easily mobilized material during heavy rainfall (Fig. 4), whereas metamorphic rocks contribute coarser and more angular fragments

(Fig. 4). Such differences in sediment character influence flow behavior, travel distance, and depositional patterns in debris flows.

Steep slopes, combined with susceptible lithologies, exacerbate debris flow hazards. For example, gullies DF1 and DF2, with steep channel gradients of 32° and 40°, respectively, are underlain by weak sedimentary rocks and have recorded higher incidences of debris flows. The presence of accumulation fans downstream indicates significant deposition of debris material.

Meanwhile, gullies DF3–DF7 exhibit channel gradients of up to 33° in certain sections, with initiation zone slopes exceeding 70°. Despite being underlain by more resistant metamorphic rocks, these areas are still prone to debris flows due to structural weaknesses caused by foliation and fracturing within the rocks.

## 4.4. Land Use and Land Cover

The analysis of Land Use and Land Cover (LUC) reveals an ongoing and pronounced decline in vegetation cover,
accompanied by a notable increase in barren land. Grassland, an essential protective biome against erosion, declined from 42% in 2001 to 35% in 2022 (Fig. 03). Similarly, broadleaf forest cover dwindled from 12% to 8%, with other vegetation categories following a downward trend, indicating a persistent loss of vegetation across the landscape. In contrast, the cropland category displayed fluctuations, potentially indicating changing agricultural practices or land management strategies. Barren land witnessed a notable expansion, escalating from 15% in 2001 to 24% in 2022. This increase in barren
land has significant implications for the region's susceptibility to debris flow events, as the lack of vegetation contributes to increased surface runoff and soil erosion. This transformation highlights increased land degradation and shifts in land utilization. The built-up areas category displayed a parallel increase, emphasizing urbanization's encroachment on natural habitats. These changes in LUC are critical in understanding how land use influences the dynamics of debris flow events, with urban areas and barren land offering less resistance to flow, thereby increasing flow velocity and damage potential.
Water bodies and snow cover exhibited relatively minor fluctuations, indicating relative stability in these categories. The findings underscore the urgency of addressing the ongoing vegetation loss and increasing barren land through targeted conservation efforts, sustainable land management, and informed policy decisions.

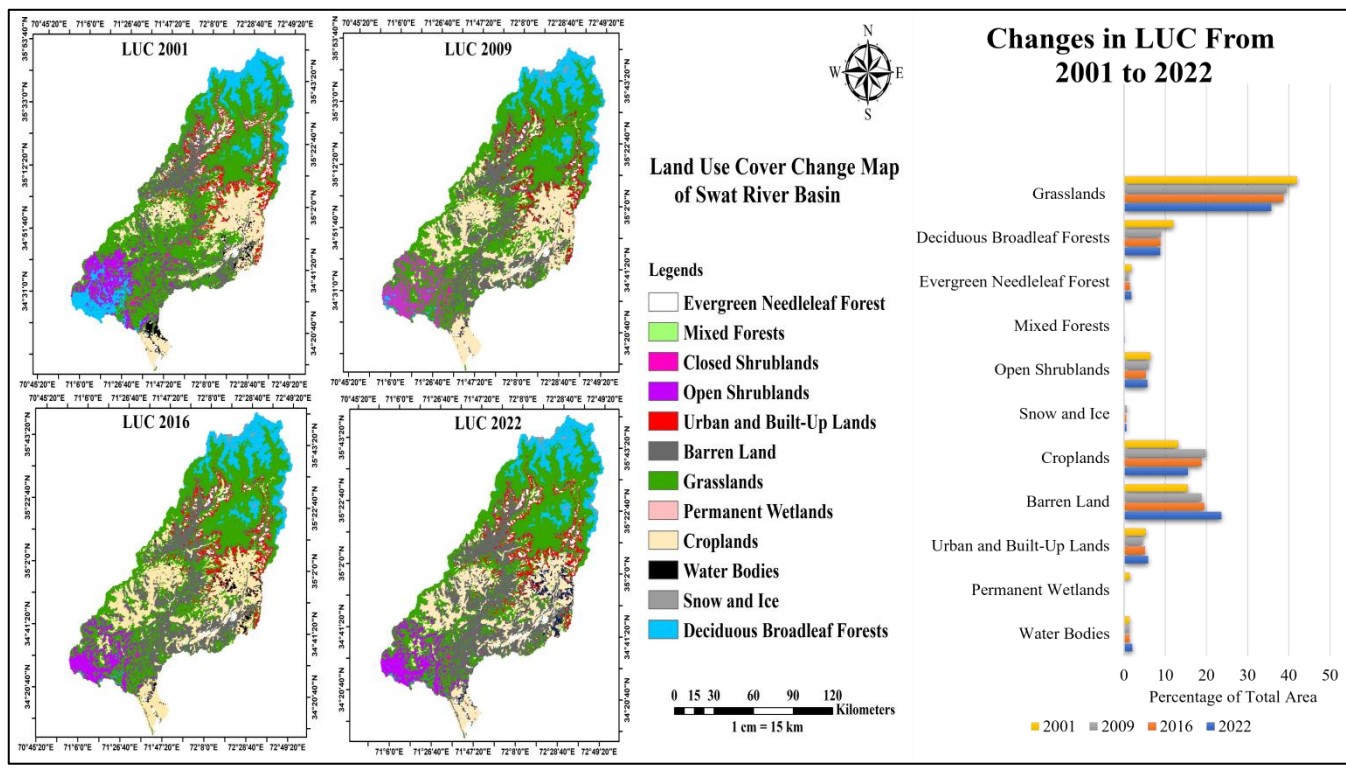

**Figure 3.** Land Use Cover Change in the Swat River Basin (2001-2022) divided into four intervals. Visualized maps are displayed on the left, while the corresponding temporal trend is presented in the right-hand panel.

### 4.5. Debris Flow fans

The extensive rainfall from August 22, 2022, to August 28, 2023, has triggered numerous debris flows in the Swat area. These debris flows deposited debris, mud, and rock along National Highway N-95 (Fig. 1), which runs parallel to the River Swat. Debris flows stand out as one of the most challenging natural events that can happen in mountainous areas, as recognized by Alexander (1991) and Hürlimann et al. (2006). Because the most significant destruction often occurs where debris flows come to rest, it is crucial to carry out detailed hazard assessments for these fan areas. This helps protect people and buildings from future debris flows and allows us to manage the associated risks better (Prochaska et al., 2008). In the surveyed region, we found 07 gullies adjacent to the River Swat that experienced debris flows during the heavy rainfall incident, as shown in Figure 4 (a). The majority of the debris flow made its way to the River Swat. These alluvial fans formed debris dams that entirely or partially obstructed the River Swat at its confluence with its tributaries. Consequently, the breakage of these unconsolidated dams resulted in widespread flooding along the River Swat. Roads, bridges, and houses suffered significant destruction due to the debris flows. The 07 debris flow catchment areas surveyed were analyzed for their morphological characteristics, summarized in Table 2 and Fig. 4. The catchment areas have a surface area ranging from 7.1 to 155 km². The identified debris flow channels were long and steep (length 5 km; average channel gradient, >30°). The catchment source areas have an internal relief that varies from 1.5 to 3 km, as shown in Table 2.

**Table 2:** Morphological characteristics of the 07 debris flow

| Gully Code | Gully Name | Basin Area Km² | Channel Length (Km) | Min Elevation (m) | Max Elev (m) | Basin Relief (m) | Channel Gradient | The slope of Initiation Zones (Degrees) | Avg Slope | Whether Debris flow has an accumulation fan (yes, no) | The shape of the accumulation fan |
|---|---|---|---|---|---|---|---|---|---|---|---|
| DF 1 | Gabral | 224 | 26 | 2222 | 4399 | 2177 | 32 | 77.8 | 10 | Yes | biased downstream |
| DF 2 | Bodai Kamar Khwar (Bahrain – Kalam Road) | 18.9 | 8.66 | 1900 | 4992 | 3092 | 40 | 77.1 | 32.1 | Yes | The river erodes symmetrical and toe. |
| DF 3 | Kulali Khwar (Bahrain – Kalam Road) | 7.1 | 5.16 | 2886 | 4418 | 1532 | 41 | 87 | 41.1 | yes | completely eroded |
| DF 4 | Chamgari Khwar (Bahrain – Kalam Road) | 26.7 | 13.5 | 3081 | 4629 | 1548 | 35 | 70.3 | 23.1 | yes | completely eroded |
| DF 5 | Daral Khwar (Bahrain) | 277 | 33.4 | 2625 | 4370 | 1745 | 33 | 77.6 | 12 | Yes | biased downstream |
| DF 6 | Madyan | 143 | 18.6 | 2122 | 4062 | 1940 | 41 | 73.3 | 17.7 | yes | biased downstream |
| DF 7 | Sakhra (Khazakhela) | 155 | 24.8 | 1972 | 3795 | 1823 | 38 | 53.9 | 11.1 | yes | biased downstream |

We conducted a rapid field measurement using handheld GPS devices and laser rangefinders to estimate the volume of the debris flow fans and the boundaries of the debris flow were established using the survey data points. The volume of debris flow estimates is shown in Fig. 4b. The debris fan volume for the smallest one was $164.063 \times 10^3$ m³, and the volume for the largest one was $501.363 \times 10^3$ m³, while the mean volume estimated was $291.658 \times 10^3$ m³. We evaluated the sizes of the deposits on the fans using the techniques recommended by Stoffel (2010).

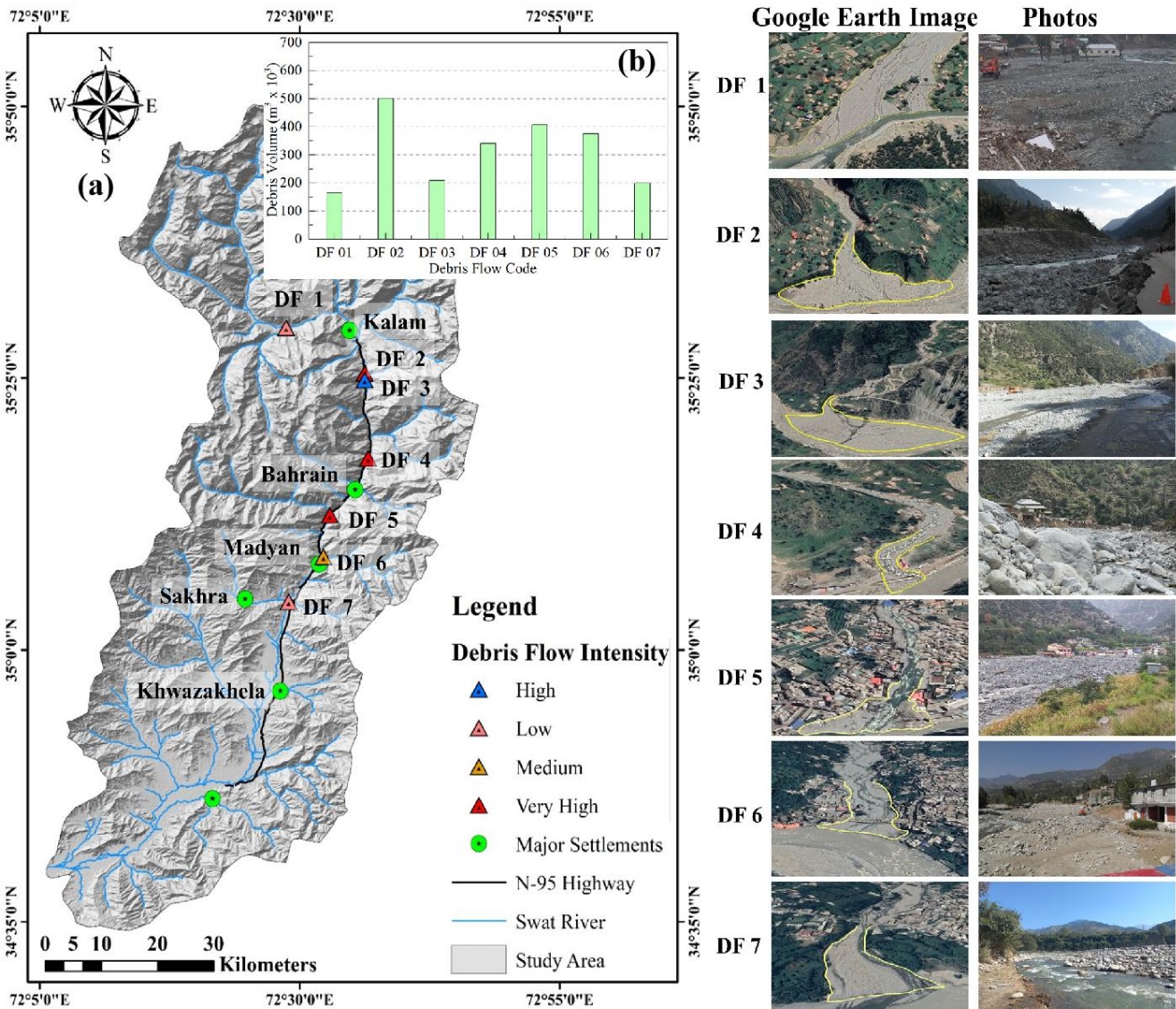

**Figure 4.** (**a**) Geographic location, catchments, and intensity of debris flows (DF 1 to 7) gullies in the Swat River basin. Debris flow intensity is categorized from low to very high. (**b**) Accumulated debris fan volume. The background image is a source of © Google Earth and field visit.

### 4.6. The triggering reason and sources for the Debris flow

Analysis of aerial photographs and on-site field surveys revealed severe ground erosion due to intense monsoon rainfall on steep terrain. Individual landslides and erosive processes are responsible for exposing bare areas on the mountainsides (Figs. 5-7). Most landslides began as relatively thin earth slides or debris slides, comprising soil mixed with rock fragments, gaining momentum as they absorbed more water and transformed into debris flows. Another way landslides began was

through the formation of small channels, known as rills, on the landslide deposits. These rills gradually deepened downstream due to erosion. When these rills formed on the landslide deposits, they funneled surface water runoff and caused it to flow in a more defined path. Consequently, this concentrated flow led to the mobilization and transportation of substantial sediment into the drainage network, resulting in the scouring of the channels (Fig. 6).

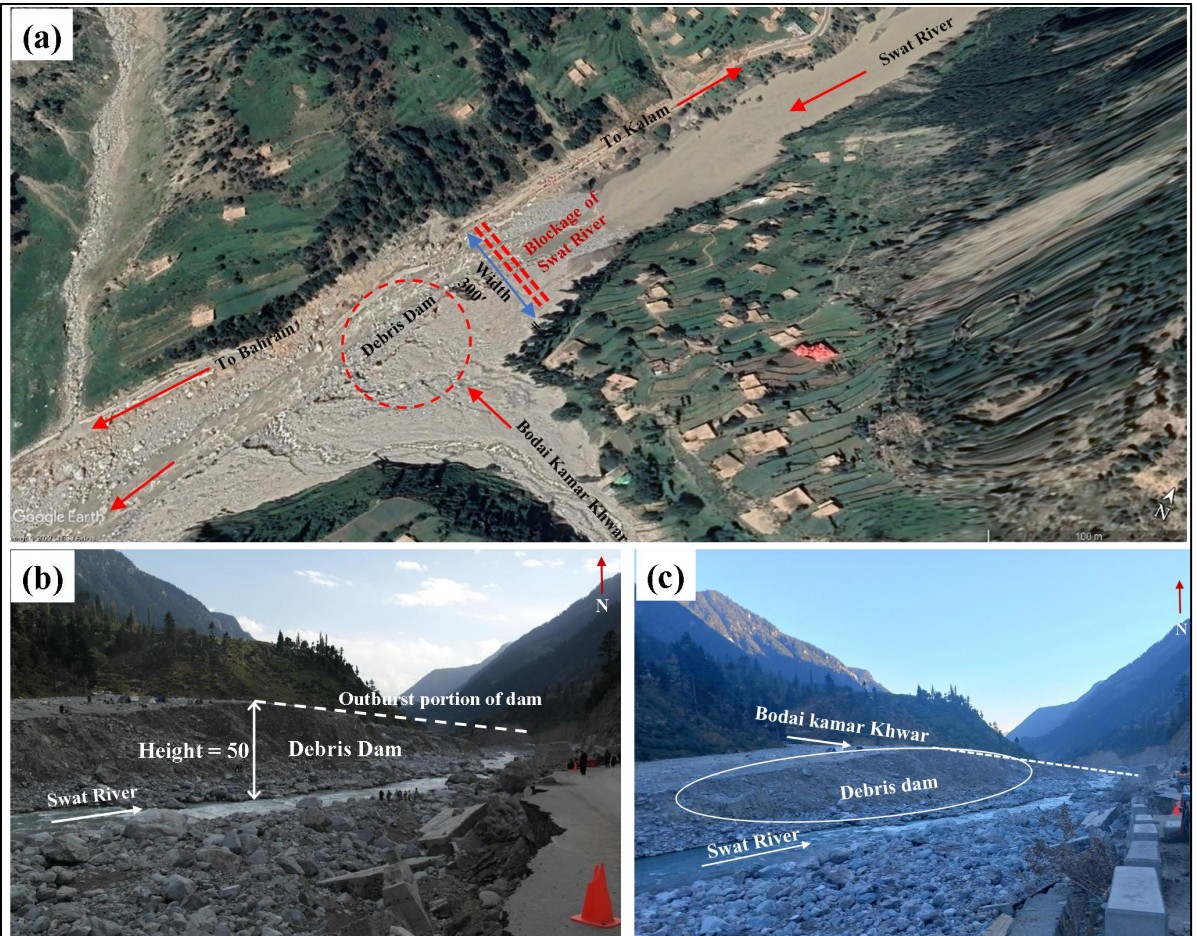

**Figure 5.** Details of the debris dam created by debris flow in Bodai Kamar gully (DF 2) (a) Aerial view of the debris dam (**b**) Frontal perspective post-failure, illustrating the height and outburst section of the debris dam (**c**) Panoramic view of the debris dam after failure. The background image for panel (**a**) is taken from © Google Earth and the other panels during the field visit.

Among the identified debris flows, DF 2 and DF 5 (Fig. 4a) were catastrophic. DF 2 originated from Budai Kamar (lat= 35°25'26.48", Long = 72°36'14.25") valley and managed to traverse the River Swat, creating a natural debris dam. This dam was approximately 50 feet in height, 500 feet long across the river, and 300 feet wide along the river section (Fig. 5). The dam blocked the River Swat for 27 hours. The breakage of the natural dam increased the discharge in the River Swat downstream of Kalam Valley, resulting in a considerable flash flow with debris in the River Swat. The stream's debris flow

was due to erosion of the stream sides, causing shallow landslides (Fig. 6) along the stream's path, leading the debris to flow with the flash flood. The general height of landslides in the stream ranged from 25 ft to 100 ft (Fig. 6). The Budai Kamar Valley (DF 2) is located on the left bank of River Swat and has a catchment area of 18.9 km² and a channel length of 8.66 km (Table 2). The catchment area of Budai Kamar Valley was determined using a 25-m DEM and on-site photos taken on 30 August 2022.

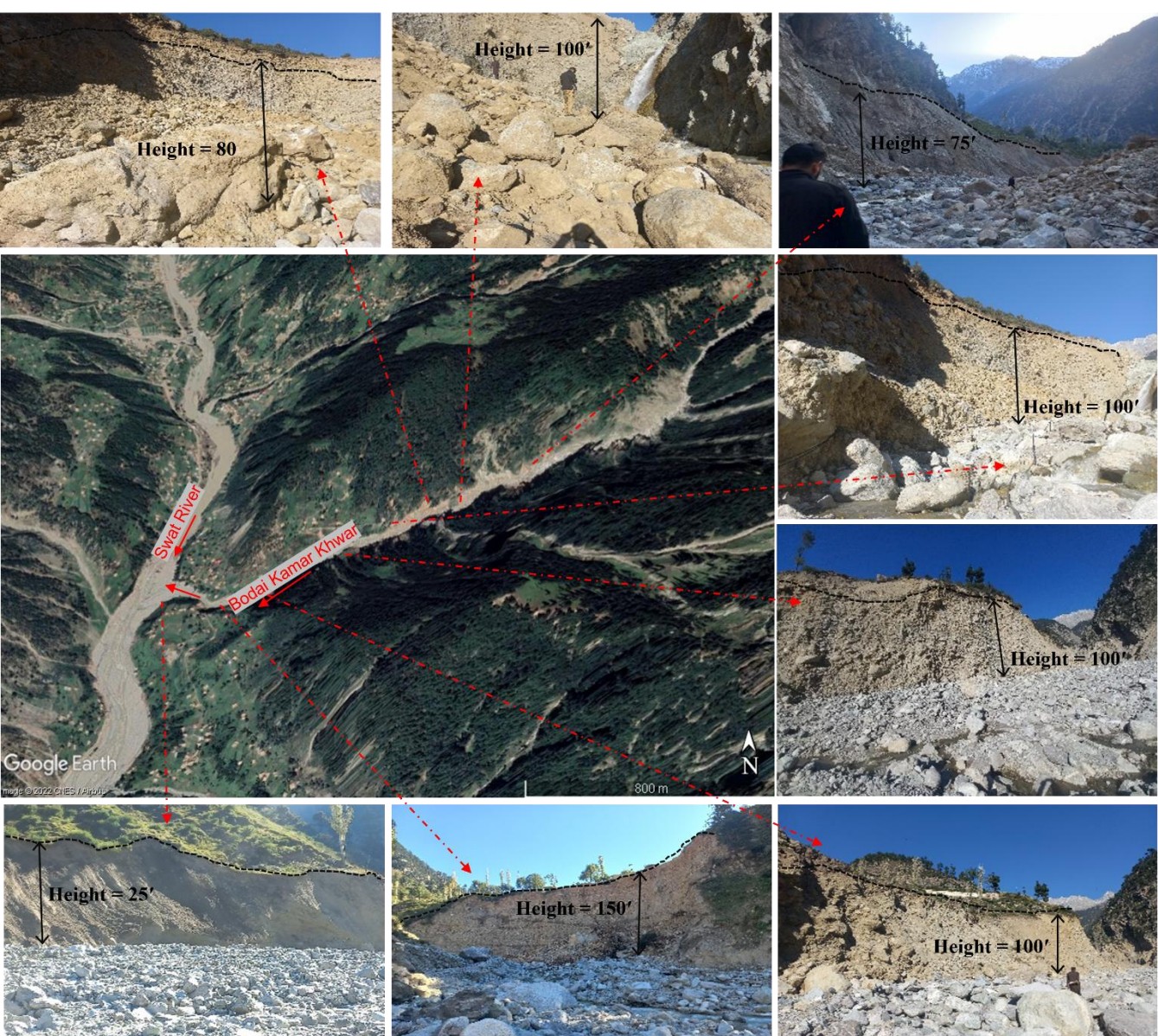

**Figure 6.** Through field investigation, identified Source areas, stream erosion, and landslide deposits contributing to debris flow events at Bodai Kamar (DF 2) locations throughout the event. The background image for Bodai Kamar Khwar is taken from © Google Earth and the other panels during the field visit.

The DF 5 originated in the Daral valley near Bahrain Bazar (Fig. 7). The Daral stream joins the River Swat near Bahrain Bazar. The extensive debris in the Daral Khwar blocked River Swat and diverted flows to Bahrain Bazar, which led to devastating damage to structures. The catchment area of the Daral Khwar is 277 km$^2$ and has a chanal length of 33.4 km. The

stream's debris flow was due to erosion of the stream sides, which caused shallow landslides (Fig. 6) along the stream's path, leading the debris to flow with the flash flood. The general height of landslides in the stream ranged from 30 ft to 70 ft (Fig 7)

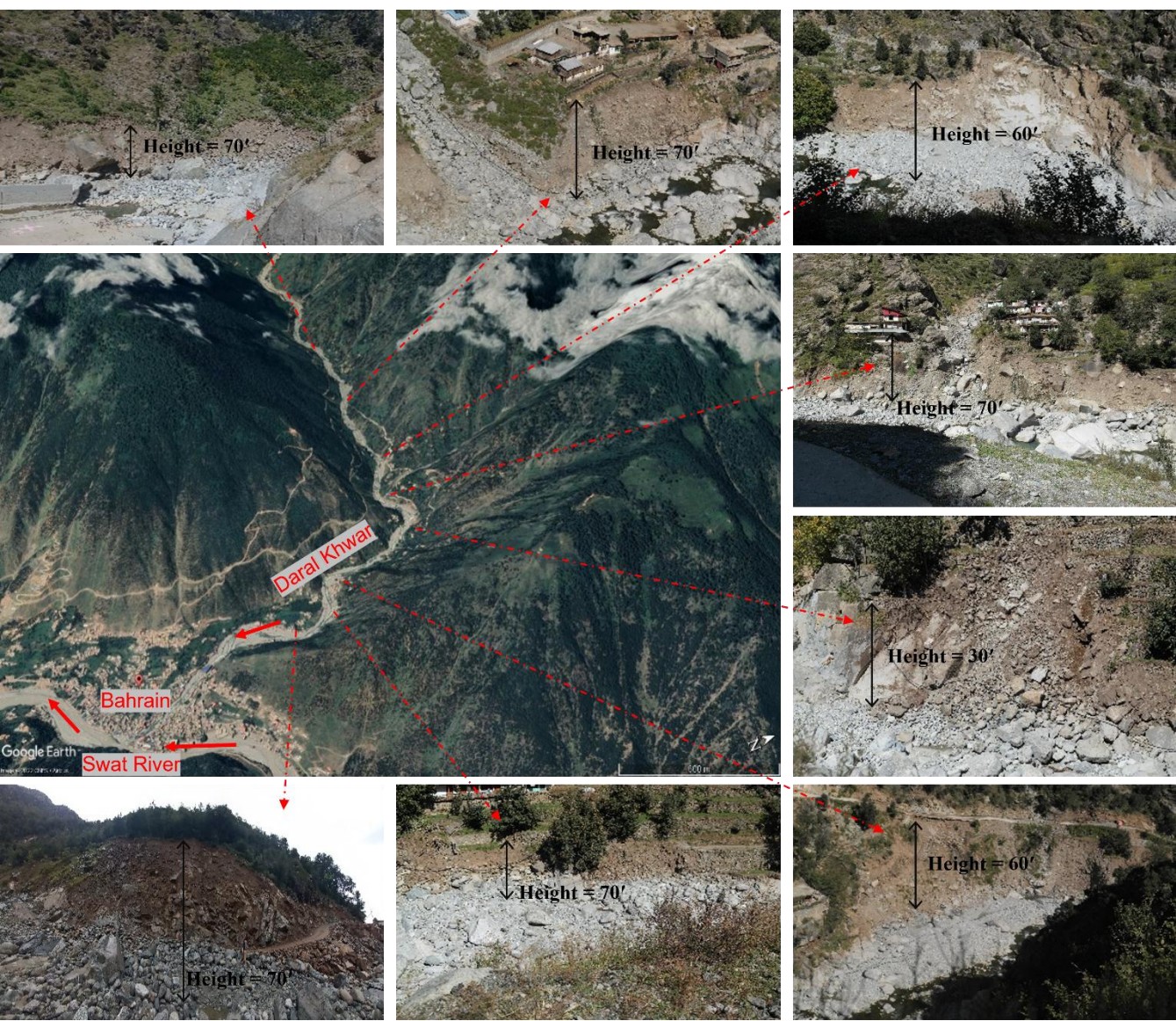

**Figure 7.** Through field investigation, identified Source areas, stream erosion, and landslide deposits contributing to debris flow events at the Daral Khwar (DF 5) location throughout the event. The background image for Daral Khwar is taken from

© Google Earth and the other panels during the field visit.

Based on our field observations of the debris flows, they appear to be triggered by substantial sheetwash and rill erosion. The intense rainfall upstream of the channel was pivotal in setting these processes in motion. As the flows advanced through the lowest parts of the gully's channel, their volume increased due to the accumulation of runoff and additional eroded material from the adjacent slopes, tributary channels, and even shallow landslides on the hillsides. Moreover, the flows also swept up larger sediment pieces stored within the channel, eventually making their way out of the catchment mouths. These more prominent elements were embedded within a finely-grained matrix, creating a mix of boulder-sized material abundant in the debris.

### 4.7. Simulated Debris flow depth and velocity

The flow depth (Fig. 8 a-e) and velocity (Fig.8 f-j) of debris flow movements triggered in four catchments (DF 2, DF 3, DF 4, and DF 5) were illustrated. The analysis covers specific time intervals: t = 0 min, 15 min, 30 min, 45 min, and 60 min. The initial conditions of the debris flow, originating from the source zone at t = 0 min before the activation of velocity from the source areas, are depicted in Fig. 8a and f. These initial conditions were significantly influenced by the prevailing Land Use and Land Cover (LUC) in the catchment areas. For example, catchments with high vegetation loss and increasing barren land exhibited more rapid acceleration of debris flow velocities and depths, aligning with findings that vegetation cover plays a critical role in mitigating debris flow dynamics (García-Ruiz, 2010). As the movements progress, at t = 15 min, the flow velocity gradually increases, reaching 8 m/s (Fig. 8 g) in the Daral Khwar streams (DB 5). Simultaneously, the flow depth at the same time is recorded as 10 to 20 m (Fig. 8 b) in the Swat River. Due to the Daral Khwar debris flow merging with the Swat River, the depth of the debris flow in the stream reaches 20-29 m. The depth and velocity of the debris flows increase rapidly as the bulbous front advances. These observed increases in flow depth and velocity can be attributed, in part, to land use changes that reduce natural barriers to flow, such as vegetation and soil cohesion. This aligns with the observations of Iverson et al. (2010), as nearly all debris flow streams merge simultaneously with the Swat River.

River.

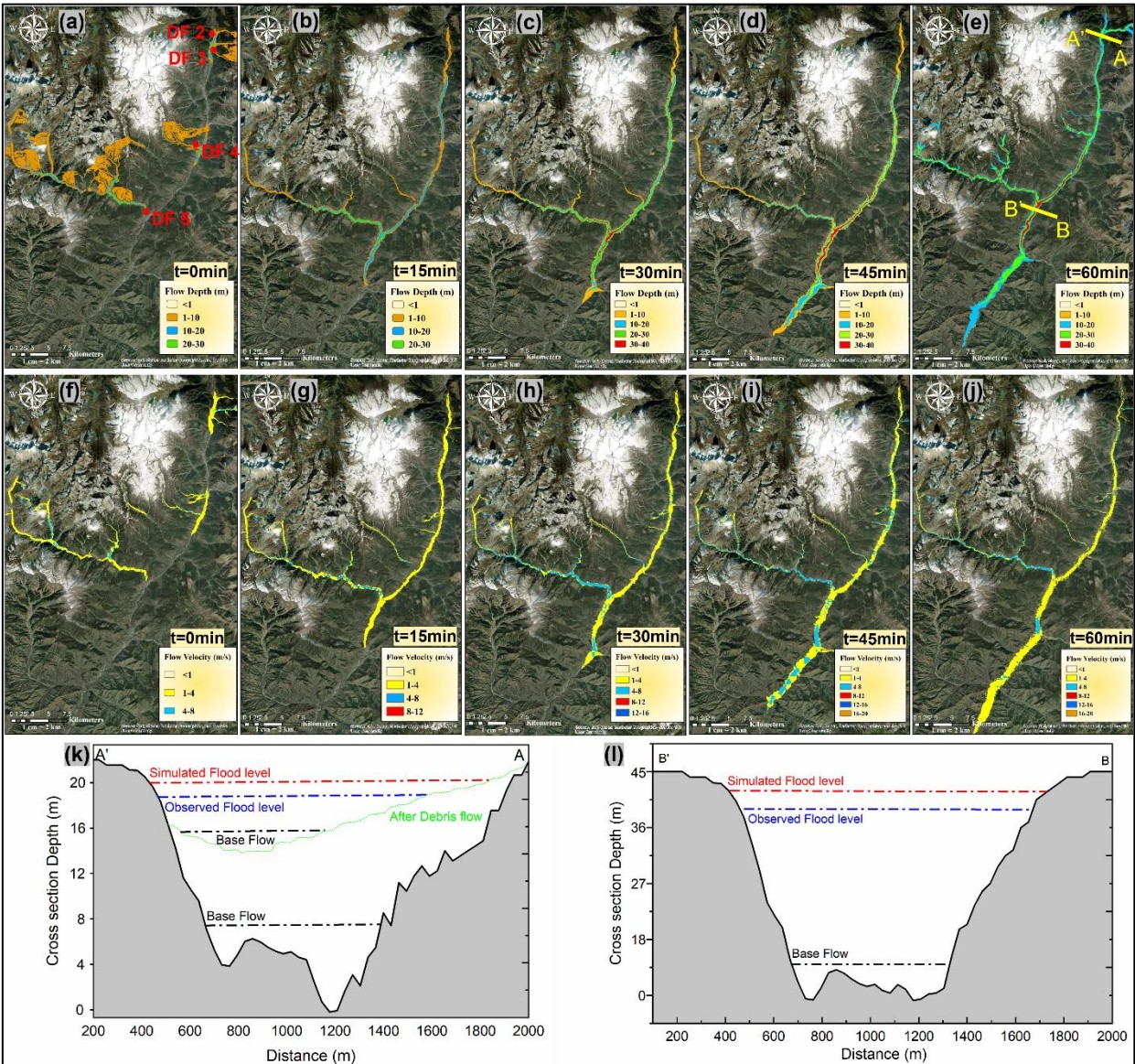

**Figure 8.** Simulated results of four impactful debris flows affecting the surrounding areas and contributing to the Swat River. Panel (**a**) shows the main debris flow and its source areas. Panels (**b-e**) illustrate the initiation and depth of the debris flows, while panels (**f-j**) showcase the corresponding velocities. Panels (**k**) and (**l**) compare the observed and simulated flood levels. Cross-sections depicted in panels (**e**) (A'A and B'B) are detailed in panels (**k**) and (**l**), explaining the observed and simulated flood levels at these cross-sections. The green horizontal dashed line in panel (**k**) shows the debris dam in the Swat River and the base flow before and after the debris flow. The background image is taken from © Google Earth.

The debris flow depth reaches its peak between 45 and 60 minutes, ranging from 36 to 40 m, while the velocity reaches a maximum of 17 to 18 m/s, as shown in panels d-e and i-j, respectively. The average peak velocity upon exiting the gullies is 12-15 m/s, and the depth of the massive material reaches 30-36 m. Regions with higher barren land and urbanized areas saw

a marked increase in the speed and magnitude of debris flows, highlighting the direct relationship between LUC and debris flow intensity. The influence and contribution of debris flow in the Swat River significantly accelerate flood flow when it enters residential areas along the river. The average velocity increases to 15 to 16 m/s, and the depth rises to 20-30 m (Fig. 8d). Panel k presents the cross-section at DF 2 (A'A), where the debris flow blocks the river. Such scenarios always recorded catastrophic downstream (Chen et al., 2024a), and the same here. Before the debris flow, the observed depth reached 18 m, and the simulated recording was 20 m. However, after the debris flows, the blockage occurs in a narrow path, holding the storage of water for a long time. These blockages suddenly breached and recorded considerable damages downstream. In Panel l, the cross-sections of DF 5 (B'B) are depicted. Due to its higher drainage area, the magnitude was high, and many sediments migrated downstream. It did not cause blockage, but considerable damages occurred due to the high magnitude along the stream, as presented in Fig. 8. The resulting floods, characterized by high velocity and magnitude, contribute to the destruction and damage of roads, houses, bridges, and farmland.

## 4.8. Impact and reason for structure damages

The catastrophic geological hazards identified in the preceding section have severely damaged the area's buildings, bridges, and roads. Most of the structural damage was caused by the direct impact of debris in the gullies. Along the River Swat, buildings were affected by flash floods triggered by the rupture of dams located at DF 2 and DF 5. The details of facilities damaged by debris and flash floods are illustrated in Fig. 8a-j. In Gabral, situated upstream of Kalam, 50 residential buildings were completely damaged due to the direct impact of debris (DF 1). Estimates from national assessments place the direct economic losses of structural damage and displacement in the Swat Valley at $2 billion, highlighting the urgent need for enhanced flood resilience in vulnerable zones. Moving to Kalam Bazar, six residential buildings suffered partial damage caused by the flooding in the Swat River, and one commercial building was washed away. At the location of Budai Kamar (DF 2), the impact of debris destroyed 42 residential buildings, while at DF 3, 20 residential buildings were also entirely damaged by the effects of debris. In Bahrain Bazar, approximately 400 commercial buildings and 9 residential buildings were damaged by the flood triggered by the rupture of the debris dam formed at DF 2. Moreover, the damages at Bahrain Bazar were further intensified by the debris at DF 5, which destroyed 45 residential buildings at Daral Stream (DF 5). At Madyan, most buildings were damaged due to the flood in the River Swat except at DF 6 and DF 7 locations. In Madyan, the flood destroyed 37 residential and 12 commercial structures, and the direct impact of debris flow damaged 20 residential buildings. The widespread damage displaced thousands of residents and disrupted essential services, impacting schooling and healthcare access for months after the floods. Rebuilding efforts in these communities have been costly and complex, impacting local economies that rely heavily on agriculture and tourism.

Structural failures in the Swat region largely result from critical design flaws that overlook the impact of debris and flash floods, especially in bridges and buildings. Field observations highlight issues like inadequate lap splice length at column bases, narrow pier spacing, low bridge clearance, and construction near riverbanks, all of which contribute to structural

vulnerability. To improve resilience, future construction should adopt flood-resilient building practices such as proper lap splice positioning, wider pier spacing, increased bridge clearance, and reinforced approaches in flood-prone areas.

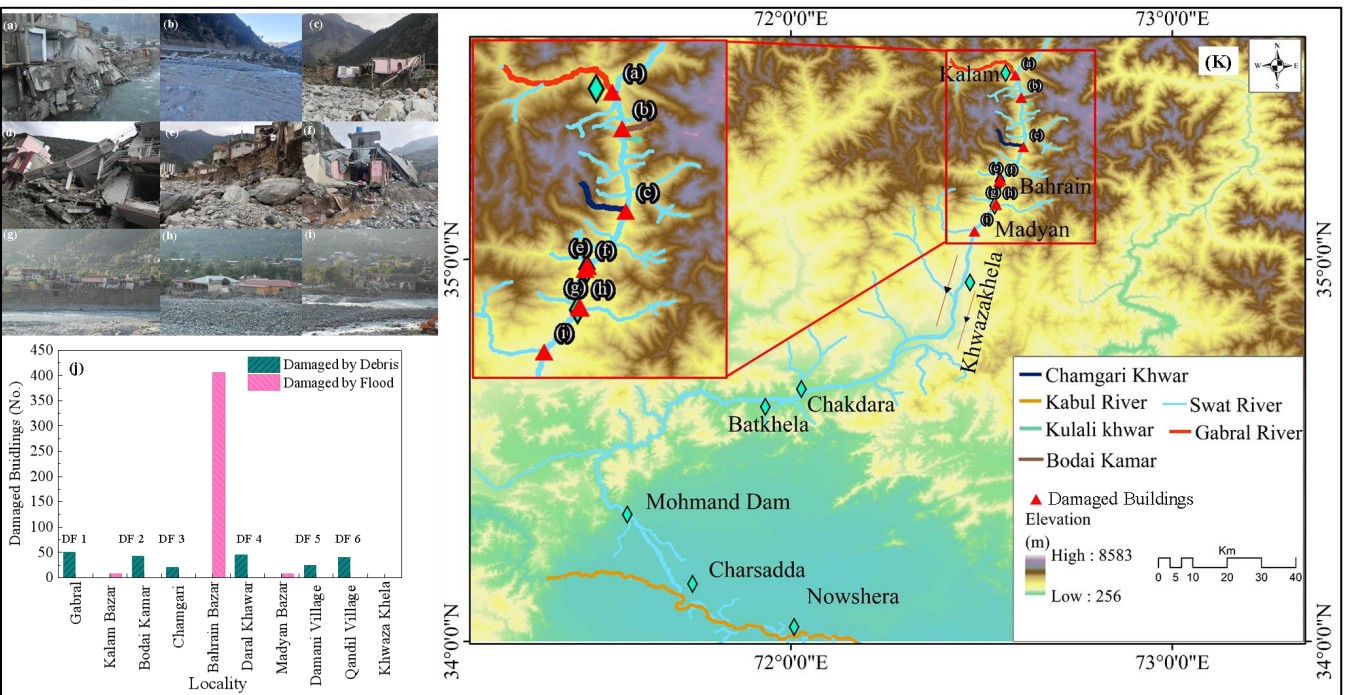

**Figure 9.** Structural Damages Unveiled: (**a**) Kalam, (**b**) Bodai Kamar Khwar (DF 2), (**c**) Chamgari Khar (DF 4), (**d**) Bahrain Bazar, (**e**) Daral Khwar (DF 5), (**f**) Near Bahrai Bazar, (**g**) Bahrain, (**h**) DF 6, (**i**) DF 7. Panel (**j**) provides a numerical breakdown of damages caused by individual debris flows and floods. The background of photos taken during a field visit.

On August 26, 2022, a total of 8 bridges were damaged by the flood along the River Swat. Out of the 8 bridges, 5 were thoroughly washed away. At the same time, 2 were partially damaged (Fig. 9). Most of the bridges collapsed due to elevated flood levels and the accumulation of debris in the River Swat. The Khatak Abad Mankyal bridge, situated upstream of Bahrain, experienced a partial collapse of its abutments and thoroughly washed away its wooden deck due to substantial debris flow and high-level pressure in the area (Fig. 9a). The direct impact of the debris flow in the gully damaged the Chamgari Bridge. The considerable debris (DF 2) blocked the bridge's opening and exerted pressure on the abutment and deck. The bridge abutment failed due to overturning, resulting in the bridge's overall failure (Fig. 9b). At the sharp curve of the river, the Kedam Bridge, located upstream of Bahrain, was entirely washed away. The bridge's low deck height, the elevated flood level, and debris caused the forceful flushing out of the bridge deck (Fig. 9c). Similarly, two bridges at Bahrain bazar (junction of River Swat and Daral Khwar (DF 5)) were washed away due to the thrust of flood and debris (Fig. 9d&e). Two bridges sustained significant damage at Madyan, which is downstream of Bahrain. The destruction resulted from the blockage of high-pressure floodwater and debris at the upstream side of the bridges, causing complete damage to the bridge approaches and retaining walls (Fig. 9f).

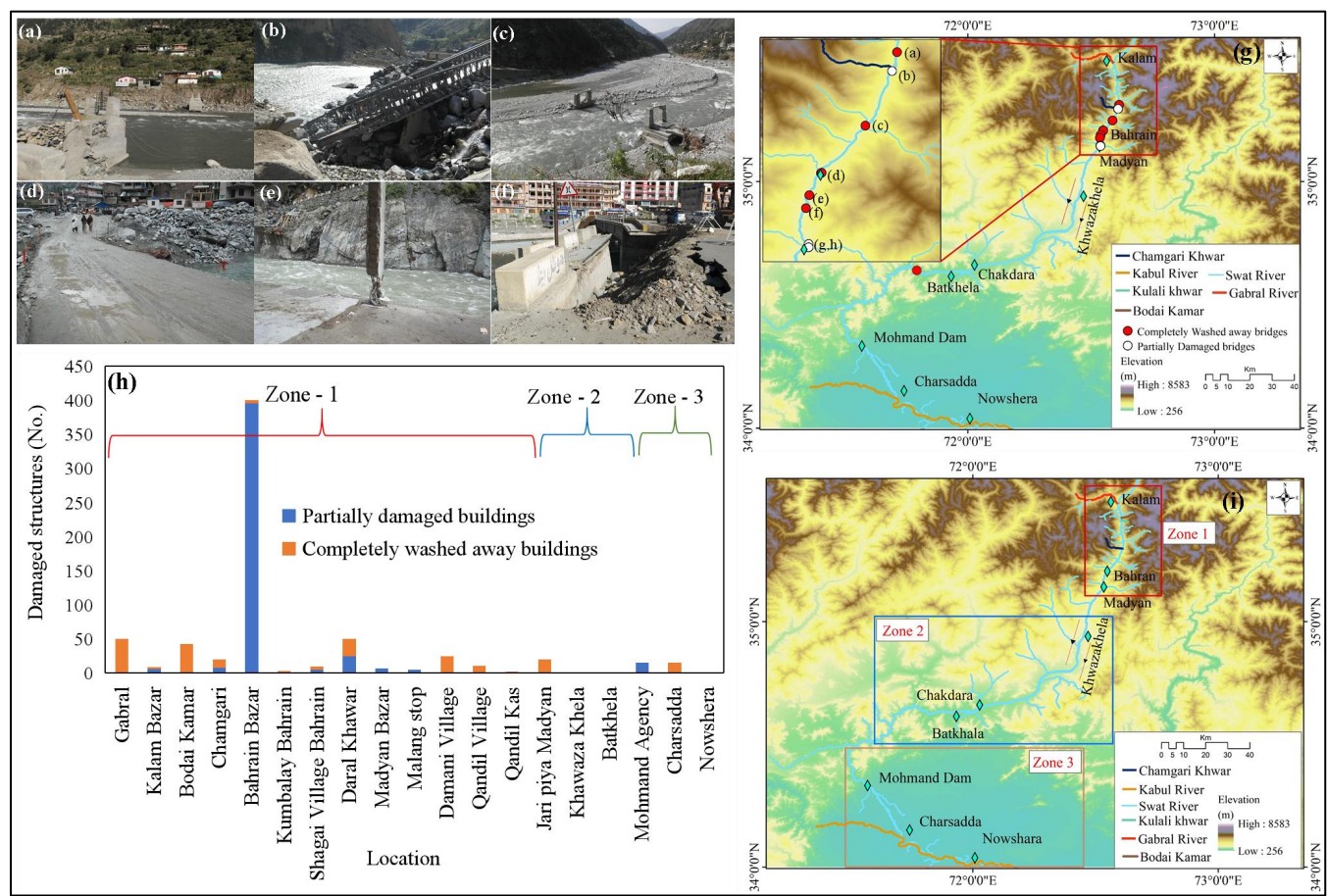

**Figure 10.** Unveiling Damaged Bridges along the Swat River: (**a**) Khatak Abad Mankyal Bridge, (**b**) Chamgari Bridge, (**c**) Kedam Bridge, (**d&e**) Bahrain Bazar Bridges, (**f**) Ayin Village Madyan Bridge, (**g&h**) Bridges at Madyan Bazar (details provided in the supplementary file). Panel (**h**) illustrates the count of wholly and partially damaged buildings in various zones within the most affected areas, as presented in panel (**i**). The background of photos of panels (**a to f**) taken during a field visit and the SRTM DEM were used for panels (**g and i**).

In addition to residents, commercial properties, and bridges, road damage along the River Swat was also observed and reported. The road was mainly damaged at the point of the river bend. The road sections were destroyed due to slope failure or retaining wall failure. In Kalam, a road spanning approximately 5 km suffered damage, while in Bahrain, a 4 km road was affected. Additionally, the road from Bahrain to Khwazakhala experienced damage of 3 km in certain sections. Below is a picture of damaged highways and their location. To better illustrate the extent of damage in regions along the Swat River, we have divided the study area into three zones (refer to Fig. 9(I)): Zone 1, Zone 2, and Zone 3. Zone 1 consists of Gabral, Kalam, Bahrain, and Madyan. Zone 2 includes Khazakhela, Chakdara, Batkhela, and Malakand Agency. Lastly, Zone 3 comprises Mohmand Agency, Charsada, and Nowshera. Zone 1 has faced severe damage to its infrastructure due to hazards and the flooding of the Swat River caused by heavy rainfall. Furthermore, the section of the river in this region is narrow, leading to high flood levels. Zone 2 encountered less damage than Zone 1 due to the broader river section in this region.

Additionally, people have become more cautious about building in flood zones after learning from the 2010 flood incident. In Zone 3, the river runs through a gorge in the Malakand Mountains until it reaches Charsada city. Moreover, after the flood in 2010, a flood dam was built in Nowshera, safeguarding the city against flood-related harm. Additionally, the destruction of bridges has severely disrupted transportation within and outside the Swat Valley, limiting access to essential resources and slowing post-disaster recovery. These disruptions highlight critical points of vulnerability in the Valley's infrastructure network.

## 5. Discussion

The findings of this study align with and build upon the broader body of literature examining the impacts of extreme weather events and geological hazards in the context of global climate change. Previous research has established that climate change exacerbates the frequency and intensity of extreme weather events, leading to increased vulnerability to natural disasters such as debris flows and floods (Masson-Delmotte et al., 2021; Tian et al., 2023). This study extends these insights by providing a detailed case study of the 2022 monsoon season in the Swat River basin, which offers a comprehensive analysis that incorporates field investigations, remote sensing data, and numerical simulations. Similar to studies by Petley (2012) and Sidle and Ochiai (2006), our research highlights the critical role of land cover changes, particularly deforestation and river encroachments, in increasing the susceptibility of regions to debris flows. As noted in our findings, the transformation of land from grasslands to barren areas reflects global trends where deforestation has been linked to increased runoff and land degradation ((Lal, 2001; Bradshaw et al., 2009). Furthermore, the rapid velocity and significant depth of debris flow observed in our numerical simulations are consistent with the mechanisms described by Hungr et al. (2002) and Choi and Liang (2024), underscoring the importance of topographical and meteorological factors in initiating and propagating debris flows. These high flow velocities are a major factor contributing to river erosion, building demolition, road and bridge damage, and loss of life in the Swat River area. By integrating these elements, our study provides a detailed and nuanced understanding of the interplay between extreme rainfall, deforestation, and topography within the context of the Swat River basin, as elaborated and discussed below.

### 5.1. The Changes in Current and Historical Trends of Monsoon Rainfall Intensity

During the 2022 monsoon season, which lasted for two months, the country experienced unprecedented rainfall, surpassing all historical records of average rainfall from 1960 to 2021. The 2022 monsoon recorded a 180% increase in rainfall compared to the country's historical average (Qamer et al., 2023). The most intense rainfall event of the season occurred in the Swat River basin between August 25 and August 28, 2022. Particularly on August 26, 2022, heavy rainfall intensity reached an astounding 71.5 mm/day, accompanied by an antecedent rainfall of an average of 40 mm over four days (see Fig. 1b). This extreme precipitation generated significant runoff and catalyzed multiple debris flows along the River Swat. August, typically considered the wettest month in 2022, witnessed an accumulative rainfall reaching an average maximum of

270 mm (Metrological Department of Pakistan). Besides meteorological factors, deforestation emerged as a crucial contributing factor to the high runoff observed during the 2022 monsoon. Evidence of deforestation was prominently depicted in Fig. 3, illustrating a notable expansion of barren land from 15% in 2001 to 24% in 2022. This transformation highlights an alarming increase in land degradation and significant shifts in land utilization (Potić et al., 2022).

The encroachment on forested areas has been a continuous trend since 2001. Notably, the affected area served as a tourist attraction and a vital source of income for local communities (Shafiq et al., 2020). The combination of increased rainfall and deforestation has significantly altered the landscape and ecological dynamics of the region (Mehmood et al., 2018; Dogan and Karpuzcu, 2022). Furthermore, deforestation has been linked to reductions in river discharge, highlighting the intricate relationship between forest cover and water resources (Stickler et al., 2013). In summary, the 2022 monsoon season was marked by unprecedented rainfall, particularly in August, with severe consequences, including debris flows. The interplay of meteorological factors and deforestation played a crucial role in exacerbating runoff and land degradation (Austin et al., 2019), emphasizing the need for sustainable land management practices to mitigate future risks.

## 5.2. The Deforestation Impact on High Runoff and Debris Flow Generation

The catastrophic events outlined in this paper serve as compelling evidence that deforestation has significantly impacted the region over the past decade, rendering it highly susceptible to more frequent occurrences of massive debris flow during the rainy season. Consequently, these debris flows transport considerably larger debris from the source area, posing a grave and substantial threat to ongoing restoration and reconstruction endeavors. Based on our investigations, it has been ascertained that the August 26 debris flow was primarily triggered by an exceptionally prolonged rainfall period spanning over four days (Fig. 3) rather than being solely influenced by an unusually high peak rainfall intensity of short duration. Prolonged rainfall can affect slope stability (Volpe et al., 2022). It has the potential to induce a saturated zone, leading to elevated pore water pressures, which, in turn, can contribute to the initiation of a landslide and its subsequent transformation into debris flows (Cojean, 1994; Chen et al., 2006; Farnsworth et al., 2019).

Deforestation is pivotal in exacerbating these geological hazards by altering the landscape's natural resilience mechanisms. Landslide deposits resulting from deforestation are often characterized by high permeability, a potentially expediting debris flow formation (Rahman et al., 2022). Given that the minimum rainfall intensity and duration required to trigger such events differ across regions, it is crucial to ascertain the prevailing conditions specific to the area under investigation in this paper. By determining these conditions, rainstorm forecasts can be utilized for early warning systems in vulnerable areas, enabling timely responses within a few hours (Campbell et al., 1989; Valente-Neto et al., 2015). The analysis of the collected field data indicates that the characteristics and spatial distribution of rainfall-induced landslides and steep topography played a crucial role in determining the locations of debris flow initiation zones. According to our findings, the runoff in the catchment areas exhibited sufficient force to mobilize a significant quantity of loose debris from the widespread landslides, ultimately transforming them into debris flows. This mechanism has been proposed by Montgomery et al. (2000) and Chen et al. (2006). The numerical simulation analysis has further elucidated that the triggered debris flow had substantial depth

and velocity. On average, the velocity increased to 18 m/ sec with an average debris depth of 40 m in 45 min. This accelerated flow significantly amplifies flooding in residential areas along the Swat River, causing substantial destruction to infrastructure and farmland. The present study has shown that rainfall-triggered landslides in different streams along the River Swat provide a tremendous volume of loose landslide debris in the debris flow source area.

## 5.3. The main reason for damages, losses, and challenges

Before this catastrophic event, people in the region were aware of debris flows but did not fully realize their potential for causing widespread and devastating impacts (Borga, 2012; Guo et al., 2021; Lijuan et al., 2017). The evaluation of debris flow hazards along River Swat had neglected mainly the interconnected effects of natural disaster chains, such as creating debris dams, dammed lakes, and floods. Thus, newly constructed infrastructure needed to be prepared and coded for the sudden and devastating impact of debris flow after a rainstorm of extreme intensity. As a result, the newly built infrastructure was ill-equipped to handle the abrupt and catastrophic effects of debris flow following an extremely intense rainstorm (Stancanelli and Musumeci, 2018). The event of 26 August 2022 indicates that debris flows are likely to develop shortly in the regions because of shallow slope failures, altering the landform and simultaneously resulting in disastrous events (Li et al., 2021). The occurrence on August 26, 2022, suggests that the areas are prone to the future development of debris flows, mainly due to shallow slope failures. These events can modify the landform and lead to disastrous consequences (Malagó et al., 2018). It is essential to highlight that identifying the areas susceptible to potential inundation by future debris flows and estimating the flow volume are necessary steps to quantify debris flow hazards effectively. These measures will facilitate appropriate land use planning to mitigate the risks posed by such events (Rodrigues et al., 2021). The risk of debris flows has significantly escalated as numerous pre-existing alluvial fans are now being utilized or considered as resettlement areas in various gullies along the River Swat.

Because many pre-existing alluvial fans are being utilized or considered as resettlement areas in the different gullies along River Swat (Islam et al., 2022a), the risk due to debris flows has dramatically increased. To mitigate debris flows in the gullies along the River Swat, it is crucial to implement not only engineering measures but also non-engineering approaches. These non-engineering measures include implementing land use zoning regulations to control and limit the utilization of hazardous areas. Additionally, relocating people residing in regions prone to debris flows and related flooding to safer locations is essential for effective mitigation.

The catastrophic event discussed in this paper has caused a severe impact on Pakistan's overall financial stability. Approximately 33 million individuals (roughly equal to 15% of the total population) were impacted. At the same time, the calamity destroyed 1.5 million residences and inflicted approximately $2.3 billion in crop damages, with infrastructural damage totalling $30 billion. Moreover, it caused extensive harm, affecting over 2000 km of roads and severing connectivity to provinces and major urban centres (Acaps, 2022; Liao, 2012). Consequently, inflation in Pakistan has surged to its peak at approximately 44.5%, signalling an impending and severe food crisis (Banking, 2022). To avoid such consequences in the future, proper measures should be taken to reduce the impact of climate change, forestation should be encouraged by federal

and local authorities, and construction in flood plans and areas susceptible to debris flows should be discouraged (Pramova et al., 2012). By prioritizing sustainable land use practices, promoting forest conservation, and implementing adequate disaster risk reduction strategies, Pakistan can enhance its resilience to monsoon-induced geological hazards and mitigate the socio-economic impacts of such catastrophic events.

## 5.4. Key Findings and Recommendations

The 2022 monsoon season in the study area brought about a harsh reality check, revealing the interconnected challenges of extreme rainfall and deforestation. To mitigate future disasters, we must adopt a comprehensive approach and models that can evaluate events in rapid responses (Gorr et al., 2022; Wang et al., 2024; De Brito and Evers, 2016). The early warning systems should be fine-tuned to account for local factors, offering timely responses to impending calamities (Nanditha et al., 2023; Nie et al., 2024). Integrating real-time satellite precipitation products and radar-based rainfall monitoring, such as data from NASA's Global Precipitation Measurement (GPM), could significantly enhance the accuracy of rainfall forecasts and early warnings. Additionally, machine learning models could analyze historical data on precipitation, land cover, and debris flow incidents, enabling more predictive, location-specific alerts. These predictive models could refine warnings by incorporating specific thresholds for rainfall and runoff, thus reducing false alarms and enhancing local responsiveness. Simultaneously, stringent land use planning and zoning regulations must be enforced to limit construction in high-risk areas, focusing on identifying regions prone to debris flows.

Long-term strategies should prioritize reforestation efforts, pivotal in stabilizing slopes, regulating water flow, and mitigating rainfall-related risks. Lastly, addressing the root causes of extreme weather events is paramount (De Brito and Evers, 2016). Intensified climate change mitigation, reduced greenhouse gas emissions, and climate-resilient agricultural practices can contribute to the region's more secure and stable future (Perkins et al., 2012; Nastos and Dalezios, 2016). This multifaceted approach, informed by the hard-learned lessons of the 2022 monsoon season, is essential to safeguard the environment and the well-being of the local communities. Furthermore, the lessons gleaned from the 2022 floods emphasize the urgency of adapting to more frequent and severe natural disasters in a warming world, highlighting the necessity for proactive climate adaptation strategies (Kamal, 2023). The severe impact of floods on public health, particularly the rise in waterborne diseases like dengue fever, underscores the critical importance of addressing health risks exacerbated by flooding events (Schmitt et al., 2023).

This study highlights the intricate relationship between deforestation and extreme climate conditions, which significantly alter mountain landscapes and trigger geological hazards, leading to severe impacts on infrastructure, GDP, and public health. Global deforestation at a rate of 10 million hectares per year and a 7-8% increase in extreme monsoon rainfall have escalated the frequency of geological hazards. From 1998 to 2017, landslides affected approximately 4.8 million people and caused over 18,000 fatalities worldwide (WHO). With rising temperatures and climate change expected to worsen these events, particularly in the Asia-Pacific region—the most disaster-prone area globally—the urgency for effective disaster management strategies is paramount. This study offers a detailed analysis of the 2022 geological hazards caused by rainfall

and deforestation, underscoring the need for reforestation and improved land use planning. By providing empirical evidence and practical insights, it calls for a redefinition of current strategies, advocating for sustainable practices to mitigate the impacts of climate change and natural hazards. This reinforces the critical need for holistic and proactive approaches in disaster management to address the complex challenges posed by these geological hazards.

In addition to short-term mitigation measures, long-term strategies are essential to build resilience against these hazards. Reforestation, especially with native or drought-resistant species, emerges as a critical solution for stabilizing soil, reducing runoff, and enhancing ecosystem health in landslide-prone areas. Integrating climate-resilient infrastructure—such as permeable pavements, bio-retention systems, and strategically placed vegetation—into land use plans can complement traditional engineering methods, providing additional stability and reducing the strain on infrastructure during extreme

weather events.

The study also underscores the importance of policy initiatives that support sustainable land use practices, particularly in vulnerable regions. These could include policies that incentivize reforestation, restrict deforestation in high-risk areas, and promote green infrastructure investments. By aligning practical, evidence-based recommendations with broader climate adaptation strategies, this study reinforces the need for holistic and proactive approaches in disaster management. Together,

these measures can address the complex challenges posed by geological hazards in the face of climate change and ensure a more resilient future for affected communities.

## 6. Conclusion

The 2022 monsoon season unfolded as an unprecedented event, crushing historical records of average rainfall dating back to 1960. Across the country, a notable 7–8% increase in rainfall was recorded compared to the historical average. However, the

645 Swat River basin experienced the season's most intense rainfall, peaking at an astonishing 71.5 mm/day on August 26, 2022, following an average of 40 mm of precipitation over the preceding four days. This extreme precipitation event triggered multiple debris flows along the Swat River, particularly in the Bodai Kamar and Daral Khwar gullies.

The geographical and morphological layout of the drainage area plays a crucial role in determining the spatial arrangement and commencement of debris flows. Examination revealed that most landslides triggered by rainfall were concentrated on

gradients exceeding 30 degrees, with 83% of these landslides transpiring at slope angles between 40 and 70 degrees. The confluence of meteorological factors and deforestation was pivotal in exacerbating runoff and land degradation during the 2022 monsoon season. The stark transformation of barren land from 15% in 2001 to 24% in 2022 illustrates the alarming increase in land degradation and shifts in land utilization. This deforestation has altered the rainfall patterns that trigger slope failures. Consequently, it can be inferred that the key elements contributing to debris flow initiation in this area are heavy

rainfall, steep catchment topography, and a substantial presence of debris. Field investigations further reveal that the debris supply primarily originates from shallow landslides, channel rill erosion on large landslides, alluvium along the channels, and colluvium and alluvium sourced from the channel banks. The field investigation also revealed that the shallow landslides

within the gullies led to the formation of small debris dams, and their subsequent rupture played a significant role in intensifying flash flooding and debris flows. Numerical simulations revealed that the debris flows reached a significantly high velocity of 18 m/sec with an average depth of 40 m, reaching this velocity and depth in 45 minutes. The continuous encroachment on forested areas, driven partly by their appeal as tourist attractions and sources of income for local communities, has further exacerbated the landscape's ecological dynamics.

The aftermath of the August 26, 2022, event underscores the region's susceptibility to future debris flows, primarily attributed to shallow slope failures. This calls for a holistic approach to debris flow risk management, encompassing stringent regulations on hazardous activities, relocating vulnerable populations, and establishing monitoring and early warning systems. Additionally, addressing the underlying causes of such extreme events, including climate change mitigation and sustainable reforestation efforts, is imperative to secure a more resilient future for the region. This multifaceted strategy, informed by the lessons of the 2022 monsoon season, is crucial for protecting the environment and the well-being of local communities. This paper pinpoints the most vulnerable regions in Pakistan.. The analysis presented here will inspire others to delve deeper into the impact of climate change, specifically focusing on temperature fluctuations, historical rainfall patterns, floods, seismic events, and more.

**Authorship Contribution Statement**

**NAB**: Conceptualization, Methodology, Software, funding acquisition, Data curation, and drafting. **MA**: Conceptualization, Methodology, Data curation, and drafting. **PC:** Conceptualization, Supervision, funding acquisition, writing-review and editing, Investigation. **APK, WH, MW, YS,** and **MR:** Conceptualization, Methodology, Software, Writing-review and editing, **TA, and LW**: Data curation, Investigation. **WH:** Data curation.

**Declaration of competing interest**

The authors declare that they have no known competing financial interests or personal relationships that could have appeared to influence the work reported in this paper.

**Disclaimer.**

Publisher's note: Copernicus Publications remains neutral with regard to jurisdictional claims made in the text, published maps, institutional affiliations, or any other geographical representation in this paper. While Copernicus Publications makes every effort to include appropriate place names, the final responsibility lies with the authors.

**Acknowledgments**

We thank the Department of Irrigation and Water Resources, NDMA, and PMD for their support in providing the necessary data. Special acknowledgment should be expressed to the China-Pakistan Joint Research Center on Earth Sciences, Islamabad, Pakistan.

**Financial support**

This study was financially supported by the National Natural Science Foundation of China (Grant no. 42350410445), the Second Tibetan Plateau Scientific Expedition and Research Program (STEP) (No. 2019QZKK0906), and the National Natural Science Foundation of China (Grant no. 4231101214; and 42201086).

**Review statement.** We sincerely thank the anonymous referees for their valuable feedback, which has greatly improved our
manuscript.

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
