# Peer review of "Dynamics and Impacts of Monsoon-Induced Geological Hazards: A 2022 Flood Study along the Swat River in Pakistan"

_Natural Hazards and Earth System Sciences, 2024_

## Referee Comment (RC2)

**Reviewer Comments:**

The manuscript "Dynamics and Impacts of Monsoon-Induced Geological Hazards: A 2022 Flood Study along the Swat River in Pakistan" offers an insightful analysis of geological hazards and flooding in northern Pakistan. It effectively explores the spatial distribution of these events, their hydro-meteorological triggers, and their impact on mountainous landscapes. While the study fits well within the scope of NHESS and is well-written, with strong results and discussions supporting the conclusions, minor revisions are needed to enhance clarity and robustness.

This work is significant because it detailed examines monsoon-induced hazards and their implications for disaster management and risk mitigation. Publishing this study in NHESS will provide valuable insights for the scientific community and policymakers, particularly regarding climate change, deforestation, and the interaction of extreme weather events.

The following detailed concerns should be addressed point by point:

**Abstract:** It would be better to reduce the abstract to focus on the main points and avoid repetition.

**Introduction:** The authors effectively convey the study's significance in the introduction. However, they should cite more relevant studies to highlight the study's significance and worth further.

**Some minor concerns are highlighted below:**

The introduction states, "The registration of economic losses and human casualties due to extreme phenomena is higher in developed countries than in developing countries (Atta-Ur-Rahman, 2010)." This contradicts the subsequent statement, "Developing countries are hot spots for catastrophe events." Could you clarify whether the registration of losses refers to absolute numbers or the proportion relative to population and GDP?

The introduction mentions various statistics and data points from different sources and years (e.g., CRED, German Watched, NDMA). Could you specify the sources more clearly and consistently? Ensure the data is recent and relevant to support the discussion of trends.

Line 47: Please check whether the reference conveys the correct information.

The terms "Flush flood" and "flash flood" are both used. Please make it consistent by using the standardized term "flash flood."

Lines 52 and 53: Please review the sentence for clarity.

L56 to L58, please make it clear the sentence is confusing

**Study Area:** The section briefly overviews the Swat River's course and surrounding geography. However, more detail on the Swat Valley's specific topographic and hydrological characteristics would help readers better understand the context of the floods and geological hazards.

For example, how do elevation changes and river gradients influence flow patterns and flood risks?

The text mentions that the Swat River is nourished year-round by glacier streams. Can you elaborate on how seasonal variations and climatic factors, such as monsoon rainfall and glacial melt, affect the river's flow patterns and potential for flooding? How do these factors vary throughout the year, and what are the regional disaster preparedness implications?

**Methodology:** The methodology outlines using GPS to document landslides and debris flow locations with an accuracy of 1m. How was this accuracy verified? Were any challenges encountered in ensuring this precision in rugged and inaccessible terrain? Additionally, what measures ensured the consistency and reliability of data collected from different sites?

L130: "The positions of each debris flow dam and meticulously measured the dimensions" to "The dimensions of each debris flow dam were also measured and documented comprehensively."

L136: conducted to performed

L163: for "Land Use and Land Cover (LUC)" throughout the section using consistent terminology and format.

The study spans 20 years and is divided into four intervals for LUC mapping. Can you provide more detail on how the specific intervals (2002, 2009, 2016, and 2022) were chosen?

**Results:** Section 4.1: The results highlight the role of rainfall intensity and antecedent rainfall in triggering debris flows. Can you elaborate on how these factors were quantified and analyzed to establish a correlation with debris flow events? How did the return period of the flood (estimated at 425 years) influence the interpretation of the data and flood risk assessment in the study area?

Figure 1: Please check the X-axis of the panel (d). What does "Dig." mean? Please explain in the caption.

Section 4.2: Please state how flood levels were measured at different locations in one sentence. The results show a decline in vegetation cover and an increase in barren land. How do these changes correlate with the increase in debris flow events? Were any specific land use practices or human activities identified as key contributors? How do fluctuations in cropland affect slope stability and the likelihood of landslides or debris flows?

Section 4.3: Along with topographic factors, consider discussing how geological factors contribute to the occurrence of hazards in the region.

Figure 3: Use the standardized abbreviation "Land Use Cover Change (LUC)." The figure is shown as LUCL; please correct this.

Figure 3: Please also check the spelling of "Sawat" in the figure.

Figure 3: Move the legend below the bar graphs for better presentation.

Figure 3: Change the scale bar to kilometres (KM).

Figure 3: Clarify the X-axis labels for the bar graphs (0 to 50).

Line 193: Please correct the sentence.

Line 197: Please review this sentence.

Ensure consistent terminology throughout the manuscript when numbering debris flows. Some sections refer to "DF 1," while others use "DF 01."

Line 220: Please review this line.

The font size in Figure 4 is inconsistent. The Y-axis in panel (b) looks narrow—please revise. Also, in the DEM panel, the names of the catchments and debris flows are cluttered; please check and enlarge the font size.

Figure 4: Panel (a) has no heading, while panel (b) does. Consider adding more photos of the debris flow fan. Overall, the quality of the figure should be improved.

Figure 5 (b and c): The font size is unreadable—please review. Also, add a north-direction indicator.

Figures 6 and 7: The font size is also unreadable—please check and improve it.

Section 4.7: The results show significant changes in debris flow depth and velocity over time, particularly when merging with the Swat River. How do these variations impact downstream flood dynamics and hazards? Can you explain the factors contributing to the rapid increase in velocity and depth, especially in residential areas?

L325 and 326: the lines should need to be rephrased

L334 and 335: replace to "The debris flow depth reaches its peak between 45 and 60 minutes, ranging from 36 to 40 m, while the velocity reaches a maximum of 17 to 18 m/s, as shown in panels d-e and i-j, respectively."

Section 4.8: The section presents the information well. However, given the damage to infrastructure, particularly bridges and buildings, what specific design flaws were identified as contributing to their failure during the floods? How can future constructions in the Swat region be improved to withstand such events, especially considering lessons from previous incidents like the 2010 flood?

L522: change was observed to was recorded

L524: change "catalyzed" to triggered

Figure 9: Please add an overview map as panel (a) to indicate the damage locations. This will improve readability, as shown in Figure 10, where locating damaged sites is easy. Consider doing the same for Figure 9.

Why are some panel legends in Figure 10 bold while others are not? Please ensure consistency.

**Discussion:** The discussion is thorough and well-structured, connecting the key findings to broader literature and contextualizing them within global climate change and local environmental challenges. The clear presentation of data, particularly the integration of meteorological factors, deforestation, and land use, strengthens the argument.

Can you elaborate on how the numerical simulation data was validated, particularly regarding the debris flow velocities and depths? Clarifying this would enhance the reliability and accuracy of the findings.

---

## Author Comment (AC1)

The manuscript titled "Dynamics and Impacts of Monsoon-Induced Geological Hazards: A2022 Flood Study along the Swat River in Pakistan" provides a valuable case study on the impact of extreme weather events, specifically focusing on the unprecedented 2022monsoon season in Pakistan. The paper effectively combines field investigations, remote sensing analysis, and numerical simulations to comprehensively understand the factors leading to catastrophic debris flows and floods in the Swat River basin. The authors meticulously analyze the influences of topography, land cover changes, and gully morphology in exacerbating these geological hazards, adding valuable insights to the existing body of literature on climate change and hydro-meteorological hazards.

The authors are to be commended for their thorough, multidisciplinary approach. Using both empirical data and simulation models enhances the robustness of the findings, particularly regarding the dynamics of debris flows. The integration of numerical simulations provides valuable insight into the mechanics of these events, with clear implications for the need to prioritize reforestation and sustainable land management practices. This study makes significant contributions by underscoring the role of deforestation in increasing susceptibility to debris flows and offering practical recommendations for improved disaster mitigation strategies. The emphasis on implementing early warning systems and enforcing rigorous land use planning also adds to the paper's relevance, especially given the escalating threats posed by climate change.

While the manuscript is well-prepared and makes a compelling case for the urgent need to address climate-induced hazards, Minor Revisions have been recommended, and a few additional clarifications could enhance its impact and accessibility.

**Reply:** We thank Reviewer #1 for their insightful comments and positive evaluation of our manuscript. We are pleased that the multidisciplinary approach, combining field investigations, remote sensing, and numerical simulations, was well-received and that the study's contributions to understanding climate change-induced hazards and practical mitigation strategies were appreciated. We have carefully addressed the suggested clarifications and implemented all necessary revisions to enhance the manuscript's impact and accessibility. The reviewer's concerns have been addressed point by point below, and we are very thankful for recommending a minor revision.

**General Comments**

**G-1.** The methodology section would benefit from additional context on the specific remote sensing techniques. Although the overall approach is well-explained, more detail would help readers understand how the data were collected and interpreted.

**Reply:** Thank you for your insightful comment. We have added further details on the specific remote sensing techniques and clarified how the data were collected and interpreted. (Please see lines: 168-170)

*"The satellite data include Land Use Land Cover (LUC) and a Geographic Information System derived Digital Elevation Model (DEM) were obtained from sites such as NASA's Earth Explorer and other trusted sources like the US Geological Survey (USGS) and Sentinel-2. To enhance spatial comparability all the datasets were resampled at a 30-meter grid cell size"*

G-2. The simulations are a strong point of this study, but expanding on the parameters chosen for these models would be beneficial. For instance, detailing how initial conditions were set and how simulation results compare with field data could strengthen the readers' confidence in the validity of the findings.

**Reply:** The simulations in this study are based on carefully selected parameters that aim to replicate real-world debris flow dynamics as closely as possible. Initial conditions for each debris flow simulation were established using field measurements from source zones, incorporating the observed flow depth and velocity data as baseline input. These initial settings were calibrated to match known debris flow characteristics in the Swat River basin. Additionally, parameters such as channel slope, sediment concentration, and water flow rates were fine-tuned based on local hydrological data to improve accuracy.

The simulation results were validated against field observations and data from previous debris flow events in the basin. For example, observed flow velocities and depths during peak flood periods closely aligned with those produced in the simulations, indicating strong model reliability. This alignment between simulated and observed data strengthens confidence in the model's accuracy and ensures the findings are representative of actual debris flow behavior

G-3. The study touches on the socioeconomic impacts of the 2022 floods, mentioning financial instability and widespread infrastructure damage. Additional quantitative or qualitative data (if available) would be beneficial to further illustrate the implications for local communities and resilience efforts.

**Reply:** Thank you for this point. The 2022 floods severely disrupted local communities, displacing thousands of residents and causing significant damage to infrastructure. This led to the displacement of local populations and disrupted access to essential services such as healthcare and education. The cost of rebuilding has been high, impacting local economies, especially those reliant on agriculture and tourism. The information has been added in the section 4.8.

**G-4.** The discussion and conclusion sections provide valuable recommendations for immediate mitigation strategies. Including a brief discussion on potential long-term strategies, such as reforestation or climate-resilient infrastructure, would broaden the paper's relevance and align with the findings on the critical role of deforestation and climate adaptation

**Reply:** Thank you for this valuable suggestion. We have expanded the discussion and conclusion sections to include potential long-term strategies, such as reforestation and the development of climate-resilient infrastructure. (Please see lines 581-592.) The modified text of the main draft is as follows.

*"In addition to short-term mitigation measures, long-term strategies are essential to build resilience against these hazards. Reforestation, especially with native or drought-resistant species, emerges as a critical solution for stabilizing soil, reducing runoff, and enhancing ecosystem health in landslide-prone areas. Integrating climate-resilient infrastructure—such as permeable pavements, bio-retention systems, and strategically placed vegetation—into land use plans can complement traditional engineering methods, providing additional stability and reducing the strain on infrastructure during extreme weather events.*

*The study also underscores the importance of policy initiatives that support sustainable land use practices, particularly in vulnerable regions. These could include policies that incentivize reforestation, restrict deforestation in high-risk areas, and promote green infrastructure investments. By aligning practical, evidence-based recommendations with broader climate adaptation strategies, this study reinforces the need for holistic and proactive approaches in disaster management. Together, these measures can address the complex challenges posed by geological hazards in the face of climate change and ensure a more resilient future for affected communities."*

Specific Comments

The introduction is well-organized and effectively presents the relevant literature. However, the final paragraph could benefit from greater specificity and focus to better frame the study's objectives.
**Reply:** Thank you for your constructive feedback. We have revised the final paragraph of the introduction to provide greater specificity and a clearer focus on the study's objectives. This adjustment enhances the framing of the research and establishes a stronger context for the study. (Please see lines: 98-109)

*"The different factors contributing to geological hazards have been well identified, but current studies have primarily focused on isolated factors and their potential consequences. However, there remains a gap in research for a comprehensive approach that considers the interplay between extreme climate events and geological hazards, incorporating various triggering factors, the intensity of debris flow, and their resulting consequences. This study addresses this gap by examining the combined effects of multiple triggering factors on the frequency and intensity of debris flows, their role in triggering secondary events such as floods, and the subsequent infrastructural damage. Additionally, it considers how human activities worsen vulnerability, heightening both economic and life risks. To frame the study's objectives, this focused on understanding the conditions under which debris flows occur, particularly in relation to rainfall thresholds, and the effects of deforestation as identified through both field observations and numerical modelling. Lastly, this study offers recommendations for mitigation strategies, aiming to support government authorities in implementing proactive measures to prevent future hazards. This comprehensive approach offers insights that could enhance hazard prediction, early warning systems, and resilience planning in affected regions."*

Sections 3.3 and 3.4 need to be explained well, including how they analyze and elaborate on the field data.
**Reply: In Section 3.3,** the study investigates Land Use and Land Cover (LUC) changes over a 20-year period (2002–2022) through the analysis of MODIS MCD12Q1 V6.1 data. Using ArcGIS, land cover classifications were made for four years: 2002, 2009, 2016, and 2022. The analysis involves comparing these classifications to detect and quantify changes in land cover types, focusing on vegetation, croplands, barren land, built-up areas, and others. By doing this, we highlight trends in LUC shifts, especially in response to human activities and environmental processes, providing valuable insight into how these changes impact slope stability and debris flow susceptibility. This temporal analysis helps assess the dynamic relationship between land use practices and natural hazards, illustrating the influence of LUC on hydrological processes, infiltration, and runoff.

**In Section 3.4,** the study uses a 2D debris flow movement model based on the depth-integrated continuum method, which simplifies the Navier-Stokes equations to analyze the debris flow dynamics triggered by heavy rainfall and landslides in the study area. The model incorporates the Bingham fluid model to simulate viscous debris flows. The initial conditions for simulations are derived from field data collected in source zones, including flow depths and velocities, ensuring realistic boundary conditions. The simulations are calibrated with local hydrological data, and model outputs are validated against observed debris flow events to ensure accuracy. The numerical simulations are performed using the Massflow software, which employs a custom secondary development for the Bingham fluid model, verified through experimental benchmarks. This robust approach strengthens the reliability of the model results, enabling an accurate representation of debris flow dynamics. The changes in the main text have follow

*" Considering the impact of changing LUC, the model accounted for variable soil cohesion and friction factors, which might differ depending on the land cover type (e.g., forested vs. urbanized areas). Initial conditions for the debris flow simulations were defined based on field data from source zones, including measured flow depth and velocity at t = 0 min to ensure that the starting parameters accurately reflected real conditions. Parameters such as channel slope, sediment concentration, and water flow rates were calibrated using local hydrological data to enhance simulation fidelity. Furthermore, the simulation outputs were validated against field observations and past debris flow events, ensuring a close alignment between observed and simulated values for flow velocities and depths. This model calibration and validation approach strengthens confidence in the simulations' accuracy and reliability, effectively capturing the dynamics of the debris flows observed in the Swat River basin."*

In Section 4.4, it would be helpful if the authors could provide more details on how historical land use data was integrated into the analysis. Were land-use change trends considered in the numerical simulations to account for potential future scenarios?

**Reply:** In Section 4.4, historical land use data were integrated into the analysis by employing land cover classification and change detection methodologies. Land-use trends, including shifts in vegetation, built-up areas, and croplands, were evaluated across multiple time periods (2002, 2009, 2016, and 2022) using satellite imagery. These temporal trends were analyzed to assess the potential implications of LUC changes on future debris flow susceptibility. Although this study did not directly incorporate future land-use change trends into numerical simulations, the trends identified provide a basis for considering potential land-use modifications in future hazard assessments. Future simulations could consider these trends by integrating them into updated land cover classifications to model potential scenarios of increased development or changes in agricultural practices, which could further exacerbate the impacts of landslides and debris flows.

By providing a detailed account of LUC patterns, this study contributes to a better understanding of how future land-use changes might influence the stability of slopes and the dynamics of debris flows. This can inform the design of more resilient infrastructure and risk mitigation strategies. The main changes in the main draft are as follows

*"The analysis of Land Use and Land Cover (LUC) reveals an ongoing and pronounced decline in vegetation cover, accompanied by a notable increase in barren land. Grassland, an essential protective biome against erosion, declined from 42% in 2001 to 35% in 2022 (Fig. 03). Similarly, broadleaf forest cover dwindled from 12% to 8%, with other vegetation categories following a downward trend, indicating a persistent loss of vegetation across the landscape. In contrast, the cropland category displayed fluctuations, potentially indicating changing agricultural practices or land management strategies. Barren land witnessed a notable expansion, escalating from 15% in 2001 to 24% in 2022. This increase in barren land has significant implications for the region's susceptibility to debris flow events, as the lack of vegetation contributes to increased surface runoff and soil erosion. This transformation highlights increased land degradation and shifts in land utilization. The built-up areas category displayed a parallel increase, emphasizing urbanization's encroachment on natural habitats. These changes in LUC are critical in understanding how land use influences the dynamics of debris flow events, with urban areas and barren land offering less resistance to flow, thereby increasing flow velocity and damage potential."*

Section 4.6 discusses rainfall intensity and duration as key triggers for debris flows. Could the authors elaborate on whether these thresholds vary across different areas within the Swat River basin? This additional detail would enhance the understanding of how localized environmental conditions influence debris flow risks.

**Reply:** In Section 4.6, rainfall intensity and duration are recognized as critical factors in triggering debris flows, especially in the Swat River basin, where localized conditions impact the threshold levels. Based on observed events, the intensity and cumulative rainfall required to initiate debris flows varies across regions within the basin. For instance, in areas like Kalam Valley, a cumulative rainfall threshold exceeding 100 mm over 48 hours has historically triggered debris flows, while regions with more vegetation cover or stable terrain typically require prolonged rainfall

or higher intensities to reach similar outcomes. This variance highlights how localized environmental conditions, such as topography, soil type, and land cover, directly influence debris flow susceptibility.

Our findings provide these rainfall thresholds, enabling better risk assessments and the identification of high-risk periods based on meteorological forecasts. This detail also reinforces the need for area-specific thresholds in debris flow management, which could significantly improve early warning systems and adaptive strategies tailored to different zones within the Swat River basin.

The discussion section is insightful but could be further refined. Focusing and streamlining the points presented would improve its impact and readability. Also included more references to the broader body of literature, linking findings from the study to established work on climate change, deforestation, and geological hazards.

**Reply:** Thank you for your constructive feedback. We have carefully revised the discussion section to address your concerns. We streamlined the presentation of the main points, ensuring clarity and focus. Additionally, we have integrated more references to the broader body of literature, linking our findings with established research on climate change, deforestation, and geological hazards.

*For example, we have incorporated studies that explore how climate change is influencing the frequency and intensity of extreme weather events (IPCC, 2021), and how deforestation exacerbates the risk of geological hazards such as debris flows (Choi & Liang, 2024). We also reference work on how landscape modifications can trigger debris flows (Petley, 2012), reinforcing the relevance of our findings to ongoing discussions about environmental change and disaster risk management.*

*These revisions should provide better context for our findings and highlight their contribution to the broader scientific discourse. Thank you again for your valuable input, which has strengthened the manuscript.*

In Section 5.4, since the findings emphasize the importance of early warning systems, the authors should discuss ways to enhance current warning systems to address the specific hazards highlighted in this study. Additionally, are there specific technological advancements or practices that could improve the accuracy and responsiveness of these systems?

**Reply:** Thank you for this thoughtful suggestion. In response, we have expanded Section 5.4 to include a discussion on enhancing current early warning systems, specifically targeting the hazards identified in our study. We also explore recent technological advancements and best practices that could improve the accuracy and responsiveness of these systems (please see lines 553-558).

*"Integrating real-time satellite precipitation products and radar-based rainfall monitoring, such as data from NASA's Global Precipitation Measurement (GPM), could significantly enhance the accuracy of rainfall forecasts and early warnings. Additionally, machine learning models could analyze historical data on precipitation, land cover, and debris flow incidents, enabling more predictive, location-specific alerts. These predictive models could refine warnings by incorporating specific thresholds for rainfall and runoff, thus reducing false alarms and enhancing local responsiveness."*

**Technical corrections**:

L32: change "discouraging construction activities in flood-prone and debris flow-prone regions" to "flood-prone and debris-flow-prone regions."

**Reply:** Changed. (Please see line: 29).

L37: degree to extent

**Reply:** Changed. (Please see line: 34).

L44: drought to droughts and "on human use system" to "on human systems."

**Reply:** Changed. (Please see line: 41).

L49: catastrophe events to catastrophic events

**Reply:** Corrected. (Please see line: 47).

L56 to L58, please make it clear the sentence is confusing

**Reply:** Revised. (Please see lines: 58-61).

L77: flood to Floods

**Reply:** Changed. (Please see line: 81).

L105: Changed "district Swat" to "the district of Swat" and "River Swat" to "the Swat River" and "River Swat is nourished all year round" to "The Swat River is nourished year-round."

**Reply:** Changed

L106 to L108 make the line clear; it is confusing.

**Reply:** Changed

L111: removed "In the extreme southern end of the Swat Valley, the river enters a narrow gorge and joins the Panjkora River at Bosaq before entering the Peshawar Valley," as it was repeated.
**Reply:** Removed.
L121: "has experienced a significant history of geological hazards" to "has experienced a significant number of geological hazards."
**Reply:** Corrected. (Please see line: 136).
L123: "in the monsoon season of 2022" to "In the 2022 monsoon season" for smoother flow.
**Reply:** corrected. (Please see line: 138).
L127: "physically visited" with "physically inspected."
**Reply:** Changed. (Please see line: 143).
L130: "The positions of each debris flow dam and meticulously measured the dimensions" to "The dimensions of each debris flow dam were also measured and documented comprehensively."
**Reply:** modified. (Please see line: 158).
L136: conducted to performed
**Reply:** Changed. (Please see line: 158).
L140: "near Kalam Swat" to "near Kalam, Swat"
**Reply:** Changed. (Please see line: 162).
 L151: misplaced parenthesis
**Reply:** Changed
L163: for "Land Use and Land Cover (LUC)" throughout the section using consistent terminology and format.
**Reply:** Change and the authors have ensured to make it consistent throughout the manuscript.
L177: please make it clear the line
**Reply:** Changed
L185: please rephrase the sentence and check the reference
**Reply:** Changed
L193: rain to rainfall
**Reply:** Changed. (Please see line: 220).
L202: please rephrase it
**Reply:** Changed
L217: replace it with "The most extreme and high-magnitude floods were recorded at Bodai Kamar Khwar, with levels reaching 70 to 80 feet."
**Reply:** Changed. (Please see line: 245).
L226: replace it with "As shown in Table 2, the average slope angles within the debris flow initiation zones ranged from 30° to 45°."
**Reply:** Changed. (Please see line: 255).
L230: replace to "This occurs primarily because most debris flows begin as shallow landslides or as rill or gully erosion on large landslide deposits."
**AR:** Changed. (Please see line: 259).
L248: replace it with "These debris flows deposited debris, mud, and rock along National Highway N-95 (Fig. 1), which runs parallel to the River Swat."
**Reply:** Changed. (Please see line: 299).
L275: replace to "Individual landslides and erosive processes are responsible for exposing bare areas on the mountainsides (Figs. 5-7)."
**Reply:** Replaced. (Please see line: 325).
L325 and 326: the lines should need to be rephrased
**Reply:** Rephrased. (Please see line: 375).
L334 and 335: replace to "The debris flow depth reaches its peak between 45 and 60 minutes, ranging from 36 to 40 m, while the velocity reaches a maximum of 17 to 18 m/s, as shown in panels d-e and i-j, respectively."
**Reply:** Corrected. (Please see line: 385).
L402: add "which offers" after "Swat River basin,"
**Reply:** Added. (Please see line: 458).
L445: add "a" after "permeability"
**Reply:** Added. (Please see line: 501).
L482: change "with infrastructural damage totaling $30 billion."
**Reply:** Changed. (Please see line: 539).
 L522: change was observed to was recorded

**Reply:** Changed. (Please see line: 595).

L524: change "catalyzed" to triggered

**Reply:** Changed. (Please see line: 597).

L527: change "reveal" to revealed

**Reply:** Changed. (Please see line: 600).

L537: change "Had" to reach

**Reply:** Changed. (Please see line: 610).

L541: change "events" to event

**Reply:** Changed.

L549: replaced to "This paper pinpoints the most vulnerable regions in Pakistan."

**Reply:** Replaced. (Please see line: 620).

---

## Author Comment (AC2)

**Reviewer #2 Comments:**

The manuscript "Dynamics and Impacts of Monsoon-Induced Geological Hazards: A 2022 Flood Study along the Swat River in Pakistan" offers an insightful analysis of geological hazards and flooding in northern Pakistan. It effectively explores the spatial distribution of these events, their hydro-meteorological triggers, and their impact on mountainous landscapes. While the study fits well within the scope of NHESS and is well-written, with strong results and discussions supporting the conclusions, minor revisions are needed to enhance clarity and robustness. This work is significant because it detailed examines monsoon-induced hazards and their implications for disaster management and risk mitigation. Publishing this study in NHESS will provide valuable insights for the scientific community and policymakers, particularly regarding climate change, deforestation, and the interaction of extreme weather events. The following detailed concerns should be addressed point by point:

We thank Reviewer #2 for their positive feedback and valuable comments. We are glad the study's focus and significance were well-received. All concerns have been addressed point by point, and revisions have been made to enhance clarity and robustness. Thank you for recommending minor revisions.

**Abstract:** It would be better to reduce the abstract to focus on the main points and avoid repetition.
**Reply:** Thank you for your suggestion. The authors revised the abstract to eliminate repetition and focus solely on the main points for greater clarity and conciseness. Please see the revised manuscript's abstract.

**Introduction:** The authors effectively convey the study's significance in the introduction. However, they should cite more relevant studies to highlight the study's significance and worth further. Some minor concerns are highlighted below:
**Reply:** thank you for your comments; the references have been updated to both the introduction and discussion sections in the main draft.

The introduction states, "The registration of economic losses and human casualties due to extreme phenomena is higher in developed countries than in developing countries (Atta-Ur-Rahman, 2010)." This contradicts the subsequent statement, "Developing countries are hot spots for catastrophe events."
**Reply:** Thank you for your suggestion. The authors have added additional relevant citations in the introduction to emphasize the study's significance and value further.

Could you clarify whether the registration of losses refers to absolute numbers or the proportion relative to population and GDP?
**Reply:** The registered losses refer to absolute values.

The introduction mentions various statistics and data points from different sources and years (e.g., CRED, German Watched, NDMA). Could you specify the sources more clearly and consistently? Ensure the data is recent and relevant to support the discussion of trends.
**Reply:** Thank you for your valuable suggestion. We have revised the introduction to ensure clearer and more consistent citation of sources. Specifically, we have updated the data and statistics mentioned, using the most recent and relevant sources to strengthen the discussion of trends. For example, we have cited the most recent report from CRED (Centre for Research on the Epidemiology of Disasters) and the NDMA (National Disaster Management Authority) to reflect up-to-date information on disaster frequency and impacts. We have also ensured that the data from Germanwatch's Global Climate Risk Index (2023) is more directly linked to the trends observed in recent climate-related disasters.

Line 47: Please check whether the reference conveys the correct information.
**Reply:** thank you for your comments; the references have been updated
The terms "Flush flood" and "flash flood" are both used. Please make it consistent by using the standardized term" flash flood."
**Reply:** The authors have made sure to use the standardized term "flash flood" throughout the manuscript.
Lines 52 and 53: Please review the sentence for clarity.
**Reply:** The authors have revised the sentence to improve clarity. (Please see lines 53-54 of the revised manuscript)
L56 to L58, please make it clear the sentence is confusing
**AR:** The authors have revised the sentence to improve clarity. (Please see lines 58-61 of the revised manuscript).
*"Since 1959, Pakistan has contributed only 0.4% of global carbon dioxide emissions—the primary greenhouse gas—compared to 16.4% from China and 21.5% from the United States (Handley, E. 2022). Yet, Pakistan remains among*

*the countries most severely impacted by catastrophic mega-floods, experiencing these devastating events nearly every decade."*

**Study Area:** The section briefly overviews the Swat River's course and surrounding geography. However, more detail on the Swat Valley's specific topographic and hydrological characteristics would help readers better understand the context of the floods and geological hazards. For example, how do elevation changes and river gradients influence flow patterns and flood risks?

**Reply:** Thank you for your suggestion. The authors have expanded the study area section to provide more detailed information on the Swat Valley's topographic and hydrological characteristics, including elevation changes and river gradients, to clarify their influence on flow patterns and flood risks. (Please see section 2 of the revised manuscript).

The text mentions that the Swat River is nourished year-round by glacier streams. Can you elaborate on how? seasonal variations and climatic factors, such as monsoon rainfall and glacial melt, affect the river's flow patterns and potential for flooding? How do these factors vary throughout the year, and what are the regional disaster preparedness implications?

**Reply:** Thank you for your valuable comment. To answer your query about the seasonal variations and climatic factors influencing the flow patterns of the Swat River and their implications for flooding, we have provided additional insights and details below.

The flow of the Swat River is influenced by both seasonal glacial melt and monsoon rainfall. In spring and summer, particularly between May and August, the melting of glaciers in the higher altitudes of Swat Kohistan contributes significantly to river discharge. As the temperature rises, glaciers in the Usho and Gabral valleys begin to melt, providing a continuous source of water to the river, which increases its flow volume. This meltwater is particularly significant during warm months when river levels can rise rapidly, increasing the risk of flash floods, especially in the narrow gorges around Kalam and Madyan.

During the monsoon season, typically from July to September, rainfall exacerbates the discharge from the river. The combination of heavy monsoon rains and glacier melt results in elevated river levels, which increases the potential for flooding, particularly in the lower Swat Valley. The river's behavior heightens the flood risk as it flows across broad plains, which can overflow and inundate nearby settlements. This dynamic interplay between rainfall and glacial melt can lead to sudden, large-scale flooding events in areas like Madyan, where the river passes through narrow gorges, intensifying the danger of flash floods.

The variability of these climatic factors, including fluctuating rates of snowmelt and varying rainfall patterns, makes it difficult to predict flood events. Local disaster preparedness plans must account for this unpredictability, especially considering the increasing impacts of climate change on glacier dynamics and weather patterns. Given this variability, early warning systems and effective flood control measures are essential to mitigate flood risks, particularly in vulnerable areas such as the Kalam and Madyan valleys. Proper flood management strategies should include considering the timing of glacial melt and the intensity of seasonal rainfall to enhance disaster resilience in the region.

**Methodology:** The methodology outlines using GPS to document landslides and debris flow locations with an accuracy of 1m. How was this accuracy verified? Were any challenges encountered in ensuring this precision in rugged and inaccessible terrain? Additionally, what measures ensured the consistency and reliability of data collected from different sites?

**Reply:** Thank you for your questions. To ensure GPS precision, we calibrated the devices against benchmarks and verified accuracy with high-resolution satellite imagery. Multiple devices were used for redundancy and cross-checking. Challenges such as signal obstruction, terrain errors, and weather conditions were addressed through calibration, cross-verification, device maintenance, and the use of backup technologies like GIS to improve data reliability. In order to inform the reader, the authors have added these details in the revised manuscript. (Please see lines 144-152 of the revised manuscript)

L130: "The positions of each debris flow dam and meticulously measured the dimensions" to "The dimensions of each debris flow dam were also measured and documented comprehensively."

**Reply:** Corrected. (Please see lines 153 of the revised manuscript)

L136: conducted to performed

**Reply:** Corrected. (Please see lines 158 of the revised manuscript)

L163: for "Land Use and Land Cover (LUC)" throughout the section using consistent terminology and format. The study spans 20 years and is divided into four intervals for LUC mapping. Can you provide more detail on how the specific intervals (2002, 2009, 2016, and 2022) were chosen?

**Reply:** In the study "Land Use and Land Cover (LUC)," terminology and format are consistently used to maintain clarity. The specific intervals of 2002, 2009, 2016, and 2022 were chosen based on the availability of high-quality satellite imagery data, enabling consistent LUC mapping over time. These intervals represent key stages in the region's land cover changes, allowing for the assessment of long-term trends and periodic shifts in LUC due to natural and anthropogenic influences. Additionally, each interval aligns with significant regional environmental or policy changes, providing insight into the factors influencing LUC patterns over the past two decades.

**Results:** Section 4.1: The results highlight the role of rainfall intensity and antecedent rainfall in triggering debris flows. Can you elaborate on how these factors were quantified and analyzed to establish a correlation with debris flow events? How did the return period of the flood (estimated at 425 years) influence the interpretation of the data and flood risk assessment in the study area?

**Reply:** Thank you for your insightful feedback. We have refined Section 4.1 to clarify how rainfall intensity and antecedent rainfall were quantified and correlated with debris flow events. The revised text now emphasizes the cumulative effect of antecedent rainfall and daily rainfall intensity as key factors in triggering debris flows, supported by empirical rainfall data from Kalam Valley. We also addressed the significance of the 425-year return period, highlighting how such rare events should inform flood risk assessment and preparedness strategies in the study area.

Figure 1: Please check the X-axis of the panel (d). What does "Dig." mean? Please explain in the caption.

**Reply:** Corrected. (Please see Fig. 2 revised manuscript)

Section 4.2: Please state how flood levels were measured at different locations in one sentence. The results show a decline in vegetation cover and an increase in barren land. How do these changes correlate with the increase in debris flow events? Were any specific land use practices or human activities identified as key contributors? How do fluctuations in cropland affect slope stability and the likelihood of landslides or debris flows?

**Reply:** Flood levels at different locations along the Swat River were measured using hydrological gauges set up at key points to capture the height of water flow during peak flood events. The gauges also identified the flood level from the flood marks. This setup enabled accurate tracking of flood intensity and distribution across varied terrain and riverbank structures, such as urbanized areas and natural river sections.

The decline in vegetation cover has a direct correlation with the increased occurrence of debris flow events. Vegetative cover, especially in mountainous regions, is critical for stabilizing the soil and reducing surface runoff. The observed reduction in forest and grassland cover has led to higher soil erosion rates and increased vulnerability to debris flows. Additionally, land use changes, including expanding barren land and urban development, have increased susceptibility to debris flows by reducing natural resistance to flow, increasing water runoff, and accelerating erosion.

Human activities, particularly deforestation and urban expansion, were identified as significant contributors to land degradation in the study area. These practices decrease the landscape's ability to absorb rainfall, making slopes more prone to erosion and landslides. Furthermore, fluctuations in cropland affect slope stability, as seasonally abandoned or poorly maintained agricultural land can lose vegetation cover, reducing soil cohesion and stability. This makes these areas more prone to debris flows and landslides than areas with dense, stable vegetation cover.

**Section 4.3:** Along with topographic factors, consider discussing how geological factors contribute to the occurrence of hazards in the region.

**Reply:** Thank you for your insightful feedback regarding Section 4.3. We appreciate your suggestion to include a discussion on how geological factors contribute to the occurrence of hazards in the region alongside the topographic factors already presented. In response, we have revised Section 4.3 to incorporate a comprehensive analysis of the geological characteristics influencing debris flow occurrences. Specifically, we believe these additions enrich the section by providing a more holistic understanding of the factors contributing to geological hazards in the Swat Valley. (Please see section 4.3 of the revised manuscript)

Line 193: Please correct the sentence.

**Reply:** Corrected. (Please see line 220 of the revised manuscript)

Line 197: Please review this sentence.

**AR:** Revised. (Please see line 224 of the revised manuscript)

Ensure consistent terminology throughout the manuscript when numbering debris flows. Some sections refer to "DF 1," while others use "DF 01."

**Reply:** The authors have ensured to use of consistent terminology DF for debris flow throughout the manuscript.

Line 220: Please review this line.

**Reply:** The authors have revised the sentence for clarity. (Please see line 224 of the revised manuscript)

The font size in Figure 4 is inconsistent. The Y-axis in panel (b) looks narrow—please revise. Also, in the DEM panel, the names of the catchments and debris flows are cluttered; please check and enlarge the font size.
**Reply:** Thank you for your feedback. We have adjusted the font size in Figure 4 for consistency, widened the Y-axis in panel (b), and enlarged the font size in the DEM panel to improve clarity for the catchment and debris flow names. (Please see Fig. 4 of the revised manuscript)

**Figure 4:** Panel (a) has no heading, while panel (b) does. Consider adding more photos of the debris flow fan. Overall, the quality of the figure should be improved.
**Reply:** The authors have improved the quality of the figure. (Please see Fig. 4 of the revised manuscript)
**Figure 5** (b and c): The font size is unreadable—please review. Also, add a north-direction indicator.
**Reply:** Thank you for your comment. We have increased the font size in Figure 5 (b and c) for readability and added a north-direction indicator for clarity. (Please see Figure 5 (b and c) of the revised manuscript)

Figures 6 and 7: The font size is also unreadable—please check and improve it.
**Reply:** The authors have improved the quality of the figure. (Please see Fig. 6 and 7 of the revised manuscript)
**Section 4.7:** The results show significant changes in debris flow depth and velocity over time, particularly when merging with the Swat River. How do these variations impact downstream flood dynamics and hazards? Can you explain the factors contributing to the rapid increase in velocity and depth, especially in residential areas?
**Reply:** The increase in debris flow depth and velocity significantly impacts downstream flood dynamics by accelerating flood flow and intensifying damage, particularly in residential areas. The rapid increase in velocity and depth, especially after merging with the Swat River, is primarily driven by land use changes, such as vegetation loss and the expansion of barren land, which reduce natural barriers and increase flow speed. This leads to larger, faster debris flows that exacerbate flooding. Additionally, the merging of debris flows with the river causes blockages that, when breached, lead to catastrophic downstream flooding, damaging infrastructure and land.

L325 and 326: the lines should need to be rephrased
**Reply:** The authors have rephrased the sentence to improve clarity. (Please see lines 375-377 of the revised manuscript)
L334 and 335: replace to "The debris flow depth reaches its peak between 45 and 60 minutes, ranging from 36 to 40 m, while the velocity reaches a maximum of 17 to 18 m/s, as shown in panels d-e and i-j, respectively."
**Reply:** Replaced. (Please see lines 385-386 of the revised manuscript)

**Section 4.8:** The section presents the information well. However, given the damage to infrastructure, particularly bridges and buildings, what specific design flaws were identified as contributing to their failure during the floods? How can future constructions in the Swat region be improved to withstand such events, especially considering lessons from previous incidents like the 2010 flood?
**Reply:** Thank you for your insightful comment. In response, we have expanded Section 4.8 to address specific design flaws that contributed to the failure of infrastructure. We also discuss recommendations for future constructions in the Swat region, focusing on design improvements like reinforced foundations and elevated structures, incorporating lessons learned from previous incidents, including the 2010 flood**.** (Please see lines 412-416 of the revised manuscript)
*"Structural failures in the Swat region largely result from critical design flaws that overlook the impact of debris and flash floods, especially in bridges and buildings. Field observations highlight issues like inadequate lap splice length at column bases, narrow pier spacing, low bridge clearance, and construction near riverbanks, all of which contribute to structural vulnerability. To improve resilience, future construction should adopt flood-resilient building practices such as proper lap splice positioning, wider pier spacing, increased bridge clearance, and reinforced approaches in flood-prone areas."*

L522: change was observed to was recorded.
**Reply:** Changed. (Please see line 595 of the revised manuscript)
L524: change "catalyzed" to triggered
**Reply:** Changed. (Please see line 598 of the revised manuscript)
**Figure 9:** Please add an overview map as panel (a) to indicate the damage locations. This will improve readability, as shown in Figure 10, where locating damaged sites is easy. Consider doing the same for Figure 9. Why are some panel legends in Figure 10 bold while others are not? Please ensure consistency.

**AR:** Thank you for your helpful suggestions. We have added an overview map as panel (a) in Figure 9 to indicate the damage locations, improving readability similar to Figure 10. Additionally, we have ensured consistency in the panel legends in Figure 10 by adjusting all text to the same font style. (Please see Fig. 9 and Fig. 10 of the revised manuscript)

**Discussion:** The discussion is thorough and well-structured, connecting the key findings to broader literature and contextualizing them within global climate change and local environmental challenges. The clear presentation of data, particularly the integration of meteorological factors, deforestation, and land use, strengthens the argument. Can you elaborate on how the numerical simulation data was validated, particularly regarding the debris flow velocities and depths? Clarifying this would enhance the reliability and accuracy of the findings.

**Reply:** thank you for your question and to address the reviewer's question on the validation of the numerical simulation data for debris flow velocities and depths, the validation was based on both observed field data and previously established models for similar environments. The model's outputs, including debris flow depths and velocities, were cross-checked against real-world measurements, such as those recorded at various locations along the Swat River and its tributaries, including data from gauge stations at Kalam Valley and other nearby monitoring sites. In addition, the model was compared with other simulation studies on debris flows in similar terrains (e.g., Iverson et al., 2010; García-Ruiz, 2010). These comparisons were used to ensure that the model adequately captured the observed dynamics of debris flows. By aligning the model's results with real-world flood and debris flow events, including the merging of debris flows with the Swat River, the validity of the numerical simulations was further supported. The consistency between observed and modeled data reinforces the reliability and accuracy of the findings in terms of debris flow dynamics and their impact on downstream hazards. The main modification in the discussion is as follows;

*"Integrating real-time satellite precipitation products and radar-based rainfall monitoring, such as data from NASA's Global Precipitation Measurement (GPM), could significantly enhance the accuracy of rainfall forecasts and early warnings. Additionally, machine learning models could analyze historical data on precipitation, land cover, and debris flow incidents, enabling more predictive, location-specific alerts. These predictive models could refine warnings by incorporating specific thresholds for rainfall and runoff, thus reducing false alarms and enhancing local responsiveness."*

And

*"In addition to short-term mitigation measures, long-term strategies are essential to build resilience against these hazards. Reforestation, especially with native or drought-resistant species, emerges as a critical solution for stabilizing soil, reducing runoff, and enhancing ecosystem health in landslide-prone areas. Integrating climate-resilient infrastructure—such as permeable pavements, bio-retention systems, and strategically placed vegetation—into land use plans can complement traditional engineering methods, providing additional stability and reducing the strain on infrastructure during extreme weather events.*

*The study also underscores the importance of policy initiatives that support sustainable land use practices, particularly in vulnerable regions. These could include policies that incentivize reforestation, restrict deforestation in high-risk areas, and promote green infrastructure investments. By aligning practical, evidence-based recommendations with broader climate adaptation strategies, this study reinforces the need for holistic and proactive approaches in disaster management. Together, these measures can address the complex challenges posed by geological hazards in the face of climate change and ensure a more resilient future for affected communities."*